# Focused Transformer: Contrastive Training for Context Scaling

**Szymon Tworkowski**[1,3*]   **Konrad Staniszewski**[1,3*]   **Mikołaj Pacek**[1,3*]   **Yuhuai Wu**[6†]

**Henryk Michalewski**[3,4]   **Piotr Miłoś**[1,2,5]

[1]IDEAS NCBR
[2]Institute of Mathematics, Polish Academy of Sciences
[3]University of Warsaw
[4]Google DeepMind
[5]deepsense.ai
[6]xAI

## Abstract

Large language models have an exceptional capability to incorporate new information in a contextual manner. However, the full potential of such an approach is often restrained due to a limitation in the effective context length. One solution to this issue is to endow an attention layer with access to an additional context, which comprises of (key, value) pairs. Yet, as the number of documents increases, the proportion of relevant keys to irrelevant ones decreases, leading the model to focus more on the irrelevant keys. We identify a significant challenge, dubbed the *distraction issue*, where keys linked to different semantic values might overlap, making them hard to distinguish. To tackle this problem, we introduce the *Focused Transformer* (FoT), a technique that employs a training process inspired by contrastive learning. This novel approach enhances the structure of the (key, value) space, enabling an extension of the context length. Our method allows for fine-tuning pre-existing, large-scale models to lengthen their effective context. This is demonstrated by our fine-tuning of $3B$ and $7B$ OpenLLaMA checkpoints. The resulting models, which we name LONGLLAMA[2], exhibit advancements in tasks requiring a long context. We further illustrate that our LONGLLAMA models adeptly manage a $256k$ context length for passkey retrieval.

## 1   Introduction

Language models have served as a catalyst for substantial advancements in several areas, including natural language processing [Radford et al., 2019, Brown et al., 2020], code generation [Chen et al., 2021, Li et al., 2022], quantitative reasoning [Lewkowycz et al., 2022] and theorem proving [Polu and Sutskever, 2020, Jiang et al., 2022, Mikuła et al., 2023]. One of the central challenges with language models is the effective incorporation of extensive new knowledge. The common practice of fine-tuning the model is not only resource-intensive and complex to manage, but it also does not always clearly indicate how to incorporate new knowledge. For example, fine-tuning on a text such as "Alice in Wonderland" does not equip the model to answer questions about the story itself, but rather

---

[*]Equal contribution   [†]Work done while at Google Research.

[2]We release the checkpoints and source code of LONGLLAMA 🐕 , see also our colabs.

37th Conference on Neural Information Processing Systems (NeurIPS 2023).

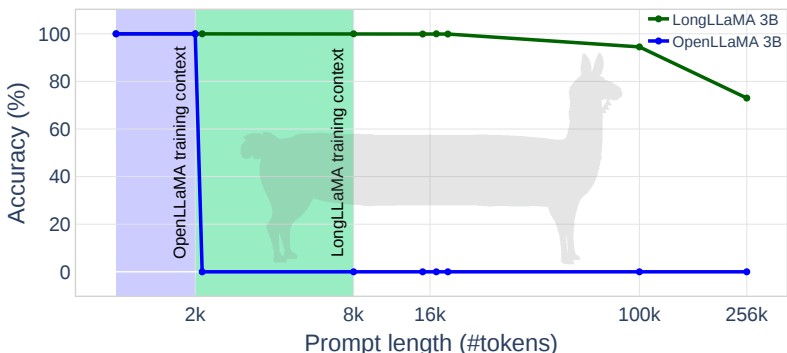

Figure 1: Accuracy of LONGLLAMA 3B on passkey retrieval compared to the original OpenLLaMA model. Our method extrapolates beyond the training length, achieving $94.5\%$ accuracy at a context length of $100k$ and $73\%$ at $256k$ tokens, while the baseline is unable to handle context longer than its training length ($2k$).

it trains the model to predict the next token or complete masked sentences. A promising alternative – integrating the new knowledge within the context – doesn't require training but is considerably restricted by the model's effective context length. For this method to work with large knowledge databases (like large code repositories), the model needs to manage a context length extending to millions of tokens.

In this research, we highlight one of the primary obstacles in augmenting the context length: as the number of documents increases, the ratio of pertinent to irrelevant tokens diminishes. The standard training procedure frequently results in overlaps between keys connected with irrelevant values and those related to relevant ones, exacerbating the model's task of differentiating between them. We term this challenge the *distraction issue*.

We propose the *Focused Transformer* (FOT), an innovative technique developed explicitly to address this issue. The Focused Transformer permits a subset of attention layers to access an additional context of (key, value) pairs through the k-nearest neighbors (kNN) algorithm, akin to the method used in [Wu et al., 2022]. This mechanism effectively extends the total context length. The distinctive aspect of the Focused Transformer is its training procedure, drawing from contrastive learning. This method addresses the distraction issue and facilitates larger context capacities. Specifically, during the training phase, we deliberately expose the chosen subset of attention layers to both relevant and irrelevant keys (like negative samples from unrelated documents). This strategy incentives the model to differentiate keys connected with semantically diverse values, thereby enhancing their structure.

We introduce and make available LONGLLAMAs (🦙), fine-tuned OpenLLaMA models with FOT, demonstrating that our method does not require long context during training and can be applied to existing models. Notably, LONGLLAMAs show significant improvements on tasks necessitating long-context modeling. In particular, they can manage a $256k$ context length on the passkey retrieval task [Mohtashami and Jaggi, 2023].

Our research contributions are the following:

**1.** We pinpoint the distraction issue as a significant challenge and a primary obstacle to scaling up the context length in Transformer models, particularly in multi-document scenarios.

**2.** We develop the Focused Transformer (FOT), designed to alleviate the distraction issue. FOT includes a unique training objective that improves the (key, value) structure, enabling the use of extensive additional context and k-nearest neighbors lookup to scale the context length.

**3.** Our method is simple to implement, and it provides the benefit of extending model context without modifying the architecture, facilitated by cost-effective fine-tuning. We demonstrate this on the $3B$ and $7B$ OpenLLaMA checkpoints. The resulting models, named LONGLLAMAs, display enhancements on tasks that benefit from increasing the number of few-shot demonstrations in the extended context, such as TREC [Li and Roth, 2002, Hovy et al., 2001] and WebQS [Berant et al.,

2013]. We also prove that for passkey retrieval Mohtashami and Jaggi [2023], our LONGLLAMA models successfully handle a $256k$ context length.

**4.** We further scrutinize FOT's capabilities across various datasets and model sizes. We show that a FOT trained with a total context of $512$ tokens can extrapolate to $16$ million tokens in a benchmark dictionary lookup task. We also assess FOT on long-context language modeling tasks such as books (PG-19), mathematics (arXiv), code (GitHub), and formal proofs (Isabelle), where it exhibits improvements in perplexity over baselines.

## 2 Related work

**Long-context transformer architectures**   A multitude of approaches have been developed to increase the context length of transformers, mostly focusing on alleviating the quadratic complexity of the attention computation. For instance, Transformer-XL [Dai et al., 2019] caches the previous context and enables the linear extension of context with the number of layers. Longformer [Beltagy et al., 2020] employs an attention mechanism that allows tokens to attend to distant tokens sparsely, reducing the computational complexity. BigBird [Zaheer et al., 2020], LongT5 [Guo et al., 2021], and [Dao et al., 2022] also use sparse attention to handle long sequences. Different efficiency considerations have been studied in [Kaddour et al., 2023], showing that they lead to limited gains. Hierarchical transformers [Nawrot et al., 2021, 2023] downsample activations in intermediate layers to reduce computation and enable longer contexts. COLT5 [Ainslie et al., 2023] proposes conditional computation to save memory and enable larger contexts. Memorizing Transformer [Wu et al., 2022] uses kNN lookup to pick up the most relevant tokens, which might also be seen as a way to reduce the computational complexity of attention. Our work adheres to this approach and aims to train a key space that handles longer attention context length (e.g., by mitigating the distraction issue) and, thus, has better long-context capabilities.

**Fine-tuning LLMs for longer retrieval**   Prior works such as RETRO [Borgeaud et al., 2022] (RETROfitting) and Memorizing Transformer [Wu et al., 2022] have demonstrated a promising path for fine-tuning existing LMs to add new capabilities without the need to retrain the entire model. In contrast to those approaches our method is not framed as a retrieval but as a way of extending the context of the model. In contrast to RETRO, we propose a single-stage method for context extension instead of a two-stage retrieve-then-embed approach. We provide a more detailed comparison with the Memorizing Transformer in Appendix C.3. More recently, a number of works have explored fine-tuning LLaMA to extend its context length. Landmark attention [Mohtashami and Jaggi, 2023] proposes a compression scheme of LLM's context into landmarks, increasing the context length of LLaMA-7B to $32K$. Position Interpolation (PI, [Chen et al., 2023] and [kaiokendev, 2023]) introduces a modification to the rotary positional encoding scheme that enables fine-tuning for $32K$ context. In contrast to this work, our method does not rely on positional encodings, following the findings from [Haviv et al., 2022]. Removing positional encoding in additional context allows us to extrapolate to $256k$ tokens, although the model was only trained on sequences up to $8K$, yielding theoretically unbounded context length.

**Zero-shot methods**   KNN-LM [Khandelwal et al., 2019] shows that one can improve the performance of a LLM by combining two probability distributions. One created by a pre-trained model, and one based on the similarity between the embedding of the currently processed token and the embeddings of tokens retrieved from a large database. Meanwhile, we extend the model context in a subset of attention layers, potentially allowing for reasoning within this extended context. Parallel Context Windows for Large Language Models [Ratner et al., 2023] introduces a method for extending the context of language models without training. They achieve this by embedding several context windows independently in parallel and allowing only a subset of tokens to attend to all windows. On the other hand, we fine-tune existing models and allow all tokens to attend to all previous tokens but only in a subset of layers. Additionally, our method allows us to improve the structure of the key-value space of the existing models.

**Contrastive learning**   Contrastive learning aims to learn good representations by comparing positive and negative examples. CLIP [Radford et al., 2021] and SimCLR [Chen et al., 2020] are two popular contrastive learning methods that have achieved state-of-the-art performance in the image domain. During contrastive pre-training, negative examples are kept in the same batch to learn to distinguish

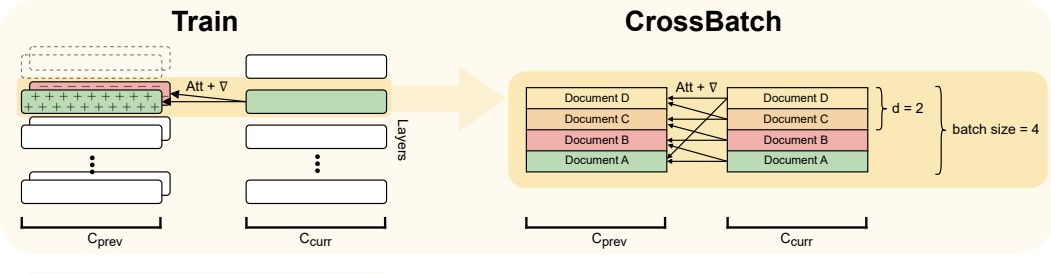

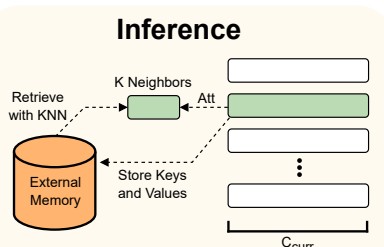

Figure 2: The Focused Transformer overview. During inference, a *memory attention layer* (green) uses additional context of $(key, value)$ pairs via kNN lookup, which effectively extends its context length. This layer is trained using *crossbatch*. Namely, the tokens from the current context $C_{curr}$ attend in a differentiable way (Att + $\nabla$) to the previous context $C_{prev}$ of the same document and, importantly, $d - 1$ contexts of other documents. The latter serve as 'negative' examples intended to better shape the $(key, value)$ space.

them from positive examples. Scaling the batch size in contrastive learning has been demonstrated to enhance the quality of representations, as shown in [Gao et al., 2021b]. It has been suggested [Gao et al., 2019] that the embedding space in language modeling suffers from degeneracy, where embeddings are tightly packed in a narrow cone, making it difficult to distinguish between them. TRIME [Zhong et al., 2022] proposes a training approach designed for training LMs with memory augmentation, which uses negatives to improve the quality of representations. The main difference between this and our approach is that we incorporate negatives into the chosen subset of attention layers instead of interpolating in the output layer and use the standard language modeling loss. TRIME [Zhong et al., 2022] also focuses on retrieval from large databases, whereas we focus on extending the context of the model. ContraCLM [Jain et al., 2023] applies contrastive losses at both the token and sequence levels during training to promote more uniformly distributed, isotropic representations. It is shown to enhance the discrimination of representations on textual semantic similarity benchmarks. While ContraCLM focuses on improving the general expressiveness of representations, our work introduces contrastive-inspired techniques designed specifically for training the attention mechanism to handle longer context lengths. Nonetheless, exploring other contrastive learning objectives could be beneficial for further improving the key structure in future work.

# 3   FOT: Focused Transformer

Our method, the Focused Transformer (FOT), is a simple plug-and-play extension of transformer models and can be used both to train new models or fine-tune existing, possibly large, models with longer context. To this end, FOT uses *memory attention layers* and the *crossbatch* training procedure. Memory attention layers enable the model to retrieve information from the additional context at inference time, effectively extending the context. The crossbatch training procedure biases the model to learn $(key, value)$ representations, which are easy to use by a memory attention layer. See Figure 2 for an overview of the FOT architecture and Appendix L for pseudocode.

## 3.1   Memory attention layers

Memory attention layers $\mathcal{L}$ are endowed with access to an additional context during inference. Namely, each query in $\ell \in \mathcal{L}$ attends to preceding keys from the local context and the top $k$ most matching keys (i.e. having the largest inner product with the query) from memory. The memory keys are ranked by the inner product with the query and retrieved using the kNN search algorithm. We use the exact kNN search implemented in FAISS [Johnson et al., 2017]. The memory is populated incrementally with $(key, value)$ pairs processed by $\ell$ beforehand. Our memory attention layer design is closely related to [Wu et al., 2022], we follow most of its design choices, except for the gating, which we replace with a simpler mechanism, which turns out to be more effective in our applications. See details in Section C.3 and Appendix B.2. We remove positional encodings in memory layers

in all our models except LONGLLAMAs. This allows LONGLLAMA checkpoints to be a drop-in replacement for LLaMA checkpoints. We treat the kNN search algorithm as an approximation of full dense attention, which opens the doors for future speed-ups.

## 3.2 Crossbatch training procedure

Our training procedure is a novel way of training (or fine-tuning) transformer-based architectures in order to improve the structure of the $(key, value)$ space. The main motivation is to shape this space so that a memory attention layer $\ell \in \mathcal{L}$ can easily focus on relevant information. The key idea, inspired by contrastive learning, is to expose $\ell$ to $(key, value)$ pairs from the current and previous local context of the given document (positives) and $d-1$ contexts from unrelated documents (negatives). Importantly, this is done in a differentiable way.

To achieve this, we use a data pipeline in which each element of the batch corresponds to a different document. We embed the previous ($C_{\mathrm{prev}}$) and the current ($C_{\mathrm{curr}}$) local context for each of the processed documents. The overview of our procedure can be found in Figure 2. Specifically for each document $\delta$ in $C_{\mathrm{curr}}$ we create a set $\{p_i^\delta\}_{i=\{1,...,d\}}$ consisting of the $(key, value)$ pairs from the previous local context of $\delta$ (positives), along with pairs from $d-1$ other contexts coming from $C_{\mathrm{prev}}$ (negatives). We also experiment with varying the number of previous contexts and negatives for different batch elements.

The operation is fully differentiable, and thus, we improve all the $(key, value)$ pairs in $p^\delta$. Two, the procedure is easy to implement; it does not require any additional loss (i.e., uses the standard transformer training objective) and is done on the level of the data loading pipeline and a minor self-attention change. The only new hyperparameter is $d$, which prescribes the ratio of positive to negative samples. Typically, we find it beneficial to start with small $d \leq 8$ (otherwise, the model tends to ignore the previous local context) and later switch to bigger values, say $d \geq 64$. Appendix B.3 provides more details about the method. Listing 1 outlines an implementation of the crossbatch.

## 3.3 The distraction issue

In this section, we conceptualize what we call the distraction issue and hypothesize it is one of the key problems in dealing with long multi-document contexts (like large code repositories). Namely, during the standard training, the model is not incentivized to distinguish the keys from different documents. We measure that the attention mass is evenly spread on the related and unrelated documents; see Figure 3. More precisely, for a document $\delta$, let $w_{ij}$ be the softmax weights related to $p_{ij}^\delta$ constructed as described in Section 3.2. We define the positive attention mass as $r_d := \sum_j w_{1j} / \sum_{i=1}^d \sum_j w_{ij}$. We observe that $r_d \approx 1/d$, which can be interpreted as the fact that the attention is equally distracted by the positive (coming from the current document at $i = 1$) and negative keys. This is an undesirable prop-

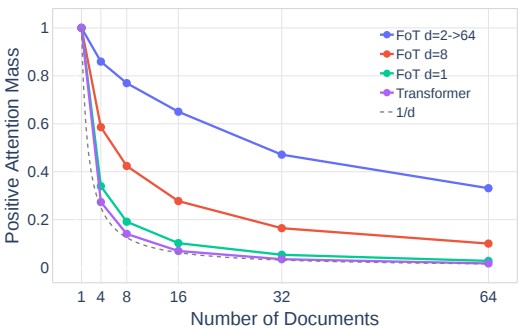

Figure 3: Distraction issue. We compare FoT trained with different values of parameter $d$ to the standard Transformer baseline. During the evaluation, both models see the previous local context and some contexts from other documents in the chosen layer (as in crossbatch training procedure). For a document $\delta$ we measure the distribution of attention mass on $p^\delta$. Scale $x$: the number of contexts from documents that the model can see. Scale $y$: avg attention mass to the previous local context of the current document.

erty since when scaling the memory, the attention becomes increasingly distracted. We show that the crossbatch mostly alleviates the distraction issue, resulting in a *focused* attention. More information can be found in Appendix B.4. In Section 5.3, we also show that the distraction issue has a harmful effect on metrics like perplexity.

# 4 LONGLLAMA 🐾: extending LLaMA's context length with FoT

One of the promises of our work is that FoT can be used to fine-tune already existing large models to extend their context length. In this section, we show that this is indeed the case. We use OpenLLaMA-3B and OpenLLaMA-7B models trained for $1T$ tokens as starting points and fine-tune them with FoT. We show that the resulting models, which we call LONGLLAMAs, are capable of extrapolating beyond their training context length (even up to $256K$) and retain the performance on short-context tasks. We release the inference code on GitHub: `https://github.com/CStanKonrad/long_llama` and the LONGLLAMA-3B checkpoint on Hugging Face: `https://huggingface.co/syzymon/long_llama_3b`. We note that our checkpoint is backward compatible, i.e. can be used with any existing LLaMA inference code (both in Hugging Face and other implementations), albeit without long-context capabilities.

## 4.1 Experimental setup

The architecture of the models is the same as OpenLLaMAs, see Geng and Liu [2023] and Appendix A.1. We use $\mathcal{L} = \{6, 12, 18\}$ (resp. $\mathcal{L} = \{8, 16, 24\}$) as the memory layers for $3B$ (resp. $7B$) LONGLLAMA model. We fine-tune the models on $10B$ (resp. $3B$) tokens using FoT, $8k$ context length and our dataset mixture based on RedPajama [TogetherComputer, 2023], see Appendix A.3.

There are three minor differences from the standard FoT procedure. First, we retain the positional encodings in the local context of the memory layers (this is not necessary for FoT, but makes our checkpoints fully compatible with any existing LLaMA inference codebase). To be more precise, queries and keys from the local context (up to $2K$ tokens) receive the standard LLaMA rotary positional encoding, whereas memory keys are encoded as if they had position 0 in the local context window. Second, we use dense attention instead of the kNN retrieval, as we found only marginal performance differences, and it is simpler to implement. Third, we modify the crossbatch training procedure to have more fine-grained control over the number of additional contexts and the ratio of positive to negative samples. All these differences are detailed in Appendix A.2.

## 4.2 Context length extrapolation on the passkey retrieval task

We first measure the effective context length of LONGLLAMA, namely the distance for which tokens can effectively attend each other. We use passkey retrieval introduced in [Mohtashami and Jaggi, 2023], a synthetic task designed to measure this property. In this task, the model has to retrieve a passkey placed randomly in a long prompt. Results are shown in Figure 1 - importantly, our $3B$ model is capable of solving this task much beyond its training context length $8K$, achieving $94.5\%$ accuracy for prompts of length $100k$ and $73\%$ for $256k$.

## 4.3 Question answering over research papers

In Table 6 we present the performance on the validation set of Qasper [Dasigi et al., 2021] from SCROLLS [Shaham et al., 2022] and compare our results to LongChat 7B [Ma and Zhang, 2023] and two baseline short-context models. We note that our model shows gains from increased context length.

## 4.4 Improving few-shot learning accuracy with longer context

We measure long-context capabilities of these models on two downstream tasks, TREC question classification [Li and Roth, 2002, Hovy et al., 2001] and WebQS question answering [Berant et al., 2013]. We follow the experimental setup of [Hao et al., 2022]. Namely, we few-shot prompt the models with as many demonstration examples as possible up to the given context length. We do not use structured prompting like in [Hao et al., 2022] - instead, we directly provide all demonstrations in context.

We observe significant accuracy gains from longer contexts on TREC and some improvements on WebQS (see Table 1). The TREC dataset consists of 50 classes. A model is tasked to predict the class label given in-context examples. Only 100 examples fit the standard context length ($2K$); it is not unusual that no class example is present for a given question, making the task impossible. Increasing the context length and the number of examples mitigates this risk. Moreover, having more demonstrations of the given class is also likely to be beneficial.

Table 1: Few-shot in-context learning performance of LONGLLAMA; accuracy on TREC and WebQS. We see significant gains from the additional context on the TREC dataset. To calculate the results, we average over 20 trials for sampling in-context demonstrations from the train set; the resulting confidence intervals for TREC and WebQS are smaller than $1\%$ and $0.1\%$, respectively.

| Dataset | TREC | | WebQS | |
|---------|------|------|-------|------|
| Context | LONGLLAMA 3B | LONGLLAMA 7B | LONGLLAMA 3B | LONGLLAMA 7B |
| $2K$ | 67.0 | 63.2 | 21.2 | 25.5 |
| $4K$ | 71.6 | 72.7 | 21.4 | 26.4 |
| $6K$ | 72.9 | 74.9 | 22.2 | 27.2 |
| $8K$ | **73.3** | **75.9** | **22.4** | **27.7** |

Table 2: Few-shot in-context learning performance comparison between standard fine-tuning on $4K$ context (baseline) and FoT fine-tuning on the same context length for $1B$ tokens. On TREC, FoT is able to utilize additional examples beyond its training context length to achieve higher accuracy at $8K$ context length, which is not possible for the baseline since its context is bounded to $4K$.

| Dataset | TREC | | WebQS | |
|---------|------|------------|----------|------------|
| Context | baseline | FoT (ours) | baseline | FoT (ours) |
| $2K$ | 52.8 | 55.6 | 20.7 | 20.8 |
| $4K$ | 57.2 | 60.9 | 18.7 | 21.0 |
| $6K$ | – | 61.7 | – | 21.2 |
| $8K$ | – | 62.5 | – | 20.7 |

### 4.5 Comparison to standard long-context fine-tuning

In this section, we compare FoT to standard long-context fine-tuning, showing that it already achieves better performance for the context length used for fine-tuning and, importantly, that it can extrapolate beyond this context length, which is not the case for the baseline.

For comparisons, we fine-tune two models, one trained with FoT and another one (baseline) with standard fine-tuning (done similarly to [MosaicML, 2023, Nijkamp et al., 2023]). In both cases, we use $3B$ models fine-tuned on $1B$ tokens using the $4K$ context length. We evaluate both models on a number of few-shot downstream tasks in the setting described in Section 4.4.

In most cases, see Table 2, we observe accuracy improvements when more few-shot demonstrations are provided in the extended context (from $2K$ used by OpenLLaMA to $4K$ used in our fine-tuning). On TREC, the gains from additional context are significant for both models, while on WebQS, the standard fine-tuning baseline does not provide any improvement from extended context. Notably, the model fine-tuned with FoT enjoys further accuracy gains when evaluated with context lengths beyond its training length ($6K$ and $8K$). This shows extrapolation capabilities of FoT, which are not present in the baseline (see e.g. Figure 1).

### 4.6 Performance on short-context tasks

Fine-tuning for longer contexts could hurt performance on the original context length ($2K$), as the training data distribution changes. We show that this is not the case for the LONGLLAMA models by evaluating them using the LM Evaluation Harness library [Gao et al., 2021a]. On most tasks, the performance is kept intact; see Appendix A.4 for details. This also confirms that LONGLLAMAs could be used as a drop-in replacement of LLaMA models as they are compatible with the original LLaMA inference code.

## 5 Analysis of FoT

In this section, we perform extensive experiments on smaller models to analyze and further validate our approach. In particular, we answer the following questions: (1) How does FoT perform when scaling the context length at inference time? (2) Can FoT be used to extend the context length of

an existing, pre-trained model? (3) How effectively can it handle distractions, and how does this capability translate to enhanced performance in long-context language modeling tasks? Moreover, we provide ablation studies of our method and additional analysis.

## 5.1 Experimental setup

**Architecture** For experiments described in this section we use decoder-only Transformer [Vaswani et al., 2017] models with 12 layers and $184M$ parameters (unless stated otherwise). Following Wu et al. [2022]; we pick $\ell = 8$ as the memory attention layer. We tune $k = 128$, the number of top keys retrieved by kNN. In most experiments, we start training with a small crossbatch dimension $d \leq 8$ and switch to $d \geq 64$ after some training. For more details about the architecture and hyperparameters, see Appendix B and Appendix E.

**Evaluation** We distinguish two evaluation settings: single-document (abbreviated to single-doc) and multi-document (abbreviated to multi-doc). The single-doc setting is typically used for evaluating models that process long contexts. Here, we clear the memory for each new document, ensuring that only the current document is available in the context. The multi-doc setting retains memory across multiple documents without resets. This scenario tests whether the model can ignore irrelevant information and focus on the relevant data, which can be useful in setups like repository-level code generation.

**Datasets** We evaluate on the following long-context language modeling datasets: PG-19 (English books), arXiv (mathematical papers), GitHub (code), and Isabelle (formal proofs). PG-19 [Rae et al., 2019] is a large dataset of English-language books published prior to 1919, sourced from the Project Gutenberg archive. This dataset is a well-established benchmark for evaluating long-context language models [Sun et al., 2021]. The arXiv dataset contains LaTeX source of papers labeled as "Mathematics" that were obtained by downloading articles through the arXiv Bulk Data Access. The token count per paper in this dataset is comparable to that of a book in PG19. For details on the remaining datasets, refer to Appendix H.

## 5.2 FoT fine-tuning and context length extrapolation

FoT is a minimal modification to the standard transformer architecture; therefore, it is possible to fine-tune existing models to endow them with a longer context length via the memory attention layer, as we already demonstrated in Section 4. In this section, we deepen this analysis (on a smaller model) by studying perplexity improvements on various datasets.

As a base model, we use a standard transformer model pre-trained for $100k$ steps with context of $1K$ tokens using the standard objective and fine-tune with the FoT objective (i.e. crossbatch). The data used for both fine-tuning and pre-training is the C4 dataset Raffel et al. [2019a] (we omit documents shorter than $2K$ tokens). The fine-tuning phase takes $10k$ steps. We use the crossbatch dimension $d = 128$ and local context of $1K$ tokens (context is $2K$ during training). We evaluate models in a *zero-shot* way on 4 language modeling datasets, which require long context: arXiv, PG-19, GitHub and Isabelle, see Section 5.1 and Appendix E for details.

In Table 3, we observe that FoT enjoys steady perplexity gains up to $64K$ tokens, although it was fine-tuned only with the $2K$ total differentiable context length. We compare the model perplexity to the following baselines: Memorizing Transformer (MT) [Wu et al., 2022] fine-tuned with the local context of $1K$ and memory size of $16K$, and Transformer-XL [Dai et al., 2019] fine-tuned with both local context and window length of $1K$. To ensure a fair comparison, all three models are fine-tuned from the same base checkpoint. When evaluated with a context of $2K$, our method achieves results on par with the Transformer-XL baseline, which has access to the previous context in all layers, unlike MT and FoT. Compared to the MT baseline, we achieve better scaling when evaluated with $64K$ context length and significantly better perplexity values. Unlike MT, our method does not require training on long sequences, which is reflected by the lower perplexities of FoT when evaluated in the zero-shot setting. For more details, see Appendix G.

We also confirm the context extrapolation abilities using a synthetic dictionary lookup task. In this task, the model is first provided with $k_i : v_i$ mappings and then asked what value is associated with a particular key. We train 37M parameter models using documents of length $512$. Figure 10 shows that

Table 3: Perplexity for different context lengths after fine-tuning a standard transformer model. The model is fine-tuned using the FoT objective (i.e., crossbatch) on C4 and evaluated zero-shot varying the context size. Transformer-XL [Dai et al., 2019] and Memorizing Transformer [Wu et al., 2022] fine-tuned in the same setting are used as baselines.

| Method | Context Length | GitHub | Isabelle | arXiv | PG-19 |
|---|---|---|---|---|---|
| FoT | $2K$ | 6.72 | 5.63 | 8.17 | 23.74 |
| | $4K$ | 5.88 | 4.93 | 7.44 | 23.25 |
| | $16K$ | 5.43 | 4.51 | 6.94 | 22.85 |
| | $64K$ | **5.32** | **4.44** | **6.81** | **22.65** |
| Transformer-XL | $2K$ | 6.85 | 5.76 | 8.21 | 23.57 |
| Memorizing Transformer | $2K$ | 8.10 | 7.34 | 9.39 | 24.03 |
| | $4K$ | 7.55 | 6.93 | 8.95 | 23.62 |
| | $16K$ | 7.27 | 6.66 | 8.66 | 23.32 |
| | $64K$ | 7.26 | 6.64 | 8.60 | 23.24 |

FoT, after 5k steps of training, can effectively utilize memory consisting of 16M tokens achieving accuracy above $92\%$. Details can be found in Appendix F.

## 5.3 Handling distractions in language modeling tasks

In this section, we measure how handling distractions in the multi-document setting helps in language modeling. We pick the PG-19 dataset [Rae et al., 2019] and measure the perplexity of the next token prediction (language modeling task) when varying the size of multi-doc memory (in this case consisting of books). Intuitively, the memory tokens corresponding to the current book might be beneficial (which is also confirmed in [Wu et al., 2022]), while the ones from the other books are unlikely to be useful and thus are distractions.

We observe, see Figure 8, that higher values of the crossbatch dimension $d$ lead to better perplexity. This aligns with the observations in Section 3.3, indicating that by mitigating the distraction issue, we experience benefits in language modeling.

Moreover, all versions of FoT are able to utilize memory and achieve much better perplexity than the standard Transformer (no memory). Unsurprisingly, perplexity increases with memory size, but we stress that this happens gracefully. In the standard variant of FoT (bold line), the perplexity increases only by $0.18$ when scaling to $> 500k$ tokens. Importantly, the perplexity of FoT is close to this of Memorizing Transformer with the single-doc memory, which we treat as a soft lower bound since it is not exposed to distractions from unrelated books.

## 5.4 Context length extrapolation in single-doc

The original motivation behind FoT is to improve the multi-doc setting performance by handling distractions. Interestingly, our method also helps to extrapolate to longer contexts, even when evaluated in the single-doc setting.

To study this, we perform FoT fine-tuning (as in Section 5.2) and evaluate the perplexity of the resulting model on the PG-19 dataset with different context lengths in the zero-shot fashion. To deepen the analysis, we introduce an additional parameter $w$ (the number of previous contexts used in cross batch training procedure). We provide results for $w = 1$ (the standard setting for FoT, that corresponds to the total differentiable context being $2 \cdot 1024$) and $w = 2$ (corresponding to the total differentiable context $3 \cdot 1024$).

We observe, see Figure 9, improvements when context grows, even far beyond the training context length, which reaffirms the hypothesis that FoT helps with extrapolation to longer contexts. Moreover, $d = 2$ is significantly better than $d = 1$. When comparing $d = 1$ and $w = 2$ to $d = 2$ and $w = 1$, we observe that the former is slightly better. This is natural, as the former has longer training context.

## 5.5 Ablations and design choices

In Appendix C we present ablations on our design choices. In particular, we note the importance of differentiability and the inclusion of negatives. We also discuss the relation to Memorizing Transformer. We note that due to the limited resources we have followed the Memorizing Transformer in the choice of memory layers.

# 6 Limitations and future work

Our research opens a few avenues for future work. We list them as well as challenges and limitations.

**Scaling up context** This is by far the most important future research direction. The challenges start from purely engineering, storing more than 16M $(key, value)$ pairs will require a distributed multi-node system. In our experiments, we use the exact kNN search, which is not scalable to large memory. Using approximate kNN search will require a lot of engineering effort, as well as careful evaluation of the impact of the approximation on the model performance.

**Scaling up crossbatch** We observed that increasing $d$ is beneficial. In our experiments, we used $d = 64$ or $d = 128$, which is the maximum value that fits into the memory of a single TPUv3/TPUv2 machine, see also Appendix I. In future work, we want to further increase $d$ as well as test on devices with bigger memory or utilize multi-node training. We also note that crossbatch increases the training cost, but only in a subset of layers.

**Exploring contrastive learning** The FOT training is inspired by rather basic contrastive learning (CL) techniques. We show that this improves the key structure so that the distraction issue is mitigated. We expect that other CL methods could be beneficial, for example, hard negative mining to utilize a larger memory during training (see [Lindgren et al., 2021]). We leave this for future work.

**Combining with other methods** Developing long-context methods is an active research field, see Section 2. We believe that some of these methods could be combined with FOT, resulting in mutually beneficial interactions.

Listing 1: Possible implementation of cross-batch. To simplify the code we assume that each document occupies two consecutive elements of the batch. A more detailed version is in Appendix L.

```
# keys from other contexts will be encoded as if they
# were at the beginning of the local context
pkey_fst = pos_encode_as_first(xk=key)

# local context keys encoded in the standard way
pquery, pkey = pos_encode(xq=query, xk=key)

# for each element of the batch we calculate indices of
# the batch that will be used in cross-batch
cross_batch_rel_ids = jnp.arange(0, -num_attentions, -1)
                        .reshape(1, -1)
batch_ids = jnp.arange(0, batch_size).reshape(-1, 1)
cross_batch_selector = cross_batch_rel_ids + batch_ids

# here we want other contexts
cross_batch_keys = pkey_fst[cross_batch_selector[:, 1:]]

# here we concatenate local context with other contexts
attention_keys =
    jnp.concatenate([pkey[:, None], cross_batch_keys], axis=1)

cb_attn_weights =
    jnp.einsum("bqhd,bckhd->bhqck",
    pquery, attention_keys, precision=precision)
```

## Acknowledgments and Disclosure of Funding

We gratefully acknowledge the TPU Research Cloud program, which was instrumental to our research by providing significant computational resources. Parts of the project were realized using the resources of Poznańskie Centrum Superkomputerowo - Sieciowe. We would also like to thank Markus Rabe for reviewing the initial manuscript and Christian Szegedy, Charles Staats, and DeLesley Hutchins for helpful discussions. We are also grateful to Xinyang Geng and Hao Liu for releasing OpenLLaMA checkpoints and the EasyLM library [Geng, 2023], allowing for training these models, which significantly accelerated our research. Piotr Milos was supported by the Polish National Science Centre grant 2019/35/O/ST6/03464. Henryk Michalewski was supported by the Polish National Science Center grant UMO-2018/29/B/ST6/02959.

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

## Broader Impact

Recent rapid developments in language models have brought a lot of new capabilities. At the same, these raised concerns about the social impact and very animated discussions in the community. Our work develops a generic technique, which in principle, could be applied to virtually any language model and thus, by extending their capabilities, exacerbate threats. We note, however, that FoT does not create any new threats. Thus, we refer to the existing body of knowledge on the broader impact of language models, see e.g. Borgeaud et al. [2022].

## A LONGLLAMA

### A.1 Architecture

OpenLLaMA [Geng and Liu, 2023] is an open-source reproduction of LLaMA [Touvron et al., 2023]. It uses a decoder-only architecture with rotary positional embeddings, and a few changes including pre-normalization with RMSNorm [Zhang and Sennrich, 2019], and SiLU activation [Elfwing et al., 2017]. A SentencePiece [Kudo and Richardson, 2018] tokenizer with 32k vocabulary size is used.

### A.2 Extending context length with FoT

**Positional encodings** To achieve backward compatibility with the original LLaMA, we retain positional encodings in the local context. The tokens outside the local context are assigned the same position as the first token in the local context.

**Dense attention to longer context** To make the implementation simpler and less dependent on external software, we resign from using kNN lookup and perform attention over the whole memory. We have found only marginal performance differences between those two approaches to memory attention.

**Crossbatch details** For the 3B LONGLLAMA model, we set $\mathcal{L} = \{6, 12, 18\}$ as the memory layers. We vary the number of additional contexts $d \in \{0, 2, 3\}$ across elements of the batch by dividing batch entries into four segments of equal size. Elements from the first segment only see local context ($d = 0$). Elements from the second segment see two additional contexts ($d = 2$), one from the same document (positive) and one from a different one (negative). Elements from the third segment see three additional contexts, two positives, and one negative. The last segment consists of elements exposed to three additional contexts coming from the same document. We abbreviate this setup as $\frac{1}{4}(0, 0), \frac{1}{4}(1, 1), \frac{1}{4}(2, 1), \frac{1}{4}(3, 0)$.

For the 7B LONGLLAMA model, we set $\mathcal{L} = \{8, 16, 24\}$ as the memory layers. Here we divide batch entries into four segments and use the following setup: $\frac{1}{4}(0, 0), \frac{1}{4}(1, 2), \frac{1}{4}(2, 5), \frac{1}{4}(3, 4)$.

**Hyperparameters** We follow the choices of OpenLLaMA with respect to most of the hyperparameters, including using the same optimizer. During fine-tuning, we use a batch size of $256K$ tokens and constant learning rate of $2e-5$, which is lower than the learning rate at the end of OpenLLaMA training ($3e-5$ after 1T tokens), and weight decay of $0.01$.

### A.3 LLaMA fine-tuning dataset

We use a mixture based on RedPajama [TogetherComputer, 2023] and The Stack [Kocetkov et al., 2022] with the following proportions of each subset:

All subsets apart from `python` are taken directly from RedPajama. For the `python` subset, we gather Python source code from The Stack and, to obtain long documents for training, concatenate files that are in the same subdirectory in random order, using a similar procedure as for the GitHub dataset in Section H. Additionally, we filter out short documents for some subsets of the original RedPajama, namely shorter than the Min. doc. length column indicates.

In case one document is too short to span across several contexts for crossbatch, then we concatenate it with the next document from the dataset.

Table 4: Proportions of RedPajama subsets for the LONGLLAMA fine-tuning mixture. For `python` subset, data from The Stack is used (see text for details). We only train on documents with length being at least Min. doc. length, if specified (otherwise we train on all documents from that subset). The horizontal line separates long-context and short-context subsets.

| Subset | Sampling proportion (%) | Min. doc. length |
|---|---|---|
| arxiv | 25 | - |
| python | 25 | 4096 |
| book | 10 | - |
| common_crawl | 29 | - |
| c4 | 5 | 1024 |
| github | 2 | 2048 |
| stackexchange | 2 | 1024 |
| wikipedia | 2 | 1024 |

## A.4 Language Model Evaluation Harness

To ensure that the performance of LONGLLAMAs has not degraded in short context scenarios, we evaluate our models on the Language Model Evaluation Harness benchmark [Gao et al., 2021a]. Table 5 compares our results with OpenLLaMA [Geng and Liu, 2023]. Similarly to the authors of OpenLLaMA, we omit CB and WSC tasks.

Table 5: Model comparison across different tasks/metrics on Language Model Evaluation Harness. The LONGLLAMA models were evaluated without context extension (i.e. as standard OpenLLaMA models). Results indicate that LONGLLAMAs maintain good performance in short context scenarios.

| Task/Metric | OpenLLaMA 3B | LongLLaMA 3B | OpenLLaMA 7B | LongLLaMA 7B |
|---|---|---|---|---|
| anli_r1/acc | 0.33 | 0.32 | 0.33 | 0.35 |
| anli_r2/acc | 0.32 | 0.33 | 0.36 | 0.37 |
| anli_r3/acc | 0.35 | 0.35 | 0.38 | 0.36 |
| arc_challenge/acc | 0.34 | 0.34 | 0.37 | 0.37 |
| arc_challenge/acc_norm | 0.37 | 0.37 | 0.38 | 0.38 |
| arc_easy/acc | 0.69 | 0.68 | 0.72 | 0.70 |
| arc_easy/acc_norm | 0.65 | 0.63 | 0.68 | 0.66 |
| boolq/acc | 0.68 | 0.68 | 0.71 | 0.71 |
| hellaswag/acc | 0.49 | 0.48 | 0.53 | 0.52 |
| hellaswag/acc_norm | 0.67 | 0.65 | 0.72 | 0.71 |
| openbookqa/acc | 0.27 | 0.28 | 0.30 | 0.30 |
| openbookqa/acc_norm | 0.40 | 0.38 | 0.40 | 0.41 |
| piqa/acc | 0.75 | 0.73 | 0.76 | 0.75 |
| piqa/acc_norm | 0.76 | 0.75 | 0.77 | 0.76 |
| record/em | 0.88 | 0.87 | 0.89 | 0.89 |
| record/f1 | 0.89 | 0.87 | 0.90 | 0.90 |
| rte/acc | 0.58 | 0.60 | 0.60 | 0.59 |
| truthfulqa_mc/mc1 | 0.22 | 0.24 | 0.23 | 0.24 |
| truthfulqa_mc/mc2 | 0.35 | 0.38 | 0.35 | 0.35 |
| wic/acc | 0.48 | 0.50 | 0.51 | 0.50 |
| winogrande/acc | 0.62 | 0.60 | 0.67 | 0.67 |
| Average score | 0.53 | 0.53 | 0.55 | 0.55 |

## A.5 Question answering over research papers

We evaluate the context utilization of our model on the validation set of Qasper [Dasigi et al., 2021] from SCROLLS [Shaham et al., 2022]. Details are in the Table 6.

Table 6: Zero-shot performance with different context lengths on the validation subset of Qasper [Dasigi et al., 2021]. We use the implementation from Language Model Evaluation Harness [Gao et al., 2021a]. In the Harness implementation, yes/no questions are evaluated separately from open questions. Observe that LONGLLAMA 3B benefits from the extended context.

| Context length | OpenLLaMA 3B | LONGLLAMA 3B | LLaMA 7B | LongChat 7B |
|:---:|:---:|:---:|:---:|:---:|
| 2K | 18.7 | 18.7 | 18.7 | 19.4 |
| 4K | - | 20.7 | - | 21.2 |
| 6K | - | 23.2 | - | 25.0 |
| 8K | - | 26.6 | - | 28.8 |

## B  Architecture

This section describes the architecture and crossbatch details for non-LLaMA-based models presented in this paper. The main differences are that for LLaMA-based models (LONGLLAMA) we maintain the positional encodings (with a slight modification detailed in A.2), do not introduce the attention temperature parameter, and replace kNN with full dense attention.

### B.1  Transformer models

For non-LLaMA-based models we use the transformer architecture introduced in [Vaswani et al., 2017] with a few standard changes. First, we use only the decoder without the encoder part. Secondly, we perform layer normalization before the input of both the attention and feed-forward modules. Additionally, we use Rotary Position Embedding [Su et al., 2021], normalize keys and queries [Henry et al., 2020], and introduce a learnable temperature parameter for each attention head.

The hyperparameters for each model size can be found in Appendix E. For training the models on PG-19, we use the standard T5 tokenizer with 32k vocabulary [Raffel et al., 2019b]. The larger models in Section 5.2 are trained with a custom SentencePiece tokenizer [Kudo and Richardson, 2018] with 64k vocabulary size.

### B.2  Memory attention layer

Memory attention layer $\ell$ is one of the transformer layers, which has access to the additional context $M$. The memory stores $(key, value)$ pairs. For each query $q$ in $\ell$, we retrieve the $k$ most matching entries from $M$ and use them to compute the attention value. More precisely, we use the kNN algorithm to pick $M_{top} := \{(key_1, value_1), \ldots, (key_k, value_k)\} \subset M$ such that $\{\langle q, key_i \rangle\}_{i=1,\ldots,k}$ are the top $k$ inner products in $M$. These are merged with the part of the local context before $q$ denoted as $C_{<q}$ and used to compute the attention value using the standard Transformer formula:

$$v := \sum_{(key,v) \in M_{top} \cup C_{<q}} s(key) \cdot v, \tag{1}$$

where $s(key)$ is the softmax score for $key$. This softmax is calculated as follows:

$$softmax \left( \left[ \frac{\langle q, key \rangle}{\tau} \right]_{key \in M_{top} \cup C_{<q}} \right),$$

where $\tau$ is a temperature parameter. In this approach, *we do not distinguish between the local context and the memory.*

Another way of integrating $M_{top}$ is via gating. In this approach, we separately compute the attention value $v_M$ for $M_{top}$ and for the local context $v_C$ (using the standard Transformer formula). Then we use a gating mechanism to combine them:

$$v := v_M \cdot g + v_C \cdot (1 - g), \quad g = \sigma(b_g),$$

where $\sigma$ is the sigmoid function and $b_g$ is a trainable bias. The gating approach was proposed in [Wu et al., 2022], see formula [Wu et al., 2022, (2)].

We found our approach, i.e. using (1), to be equally effective, see Figure 4. At the same time, (1) is simpler and does not require additional parameters. Thus, we use it in our experiments.

For kNN lookup, we use the exact kNN search implemented in FAISS [Johnson et al., 2017]. The memory attention layer does not use positional encodings. The memory is populated incrementally with $(key, value)$ pairs processed by $\ell$ beforehand. In the single-doc setting, the memory is erased after each document.

We do not use the $\tau$ parameter for LONGLLAMAs as their architecture does not normalize keys and queries. For LONGLLAMAs, we also replace the kNN search with dense attention and retain positional encodings (see Appendix A.2).

### B.3   Crossbatch training procedure

In FoT we choose a subset $\mathcal{L}$ of the attention layers for later augmentation with the memory of $(key, value)$ pairs. Let $\ell$ an attention layer from $\mathcal{L}$. During the training we expose this layer to a mixture of $(key, value)$ pairs from the current local context, $C_{\mathrm{curr}}$, and the previous local context and $d - 1$ contexts from other documents, $C_{\mathrm{prev}}$; see also Figure 2 for an illustration. We achieve this by modifying the input pipeline so that each batch index corresponds to a different document (the batch index occupied by each document is fixed from the moment we load the document till we finish processing it).

More specifically, we embed the previous and the current local context for each document in the batch. Then we use $C_{\mathrm{prev}}$ as a source of the previous local context for a given document and $d - 1$ contexts from other documents. For each element of the batch, the choices of those $d$ additional contexts are fixed. We disable positional encoding in $\ell$, as we envision it to handle global information.

To be more precise, for each document $\delta$ within the batch and query $q$ from the layer $\ell$ we create the set $p^\delta$ consisting of $(key, value)$ pairs from the previous local context of document $\delta$ along with pairs from $d - 1$ contexts gathered from $C_{\mathrm{prev}}$. The attention value for $q$ is given by

$$v := \sum_{(key, v) \in p^\delta \cup C_{\mathrm{curr}}^{\delta, <q}} s(key) \cdot v, \tag{2}$$

where $C_{\mathrm{curr}}^{\delta, <q}$ consists of $(key, value)$ pairs that preceded $q$ in its local context and $s(key)$ is the softmax score for $key$. We use softmax with learnable temperature $\tau$:

$$softmax\left( \left[ \frac{\langle q, key \rangle}{\tau} \right]_{key \in p^\delta \cup C_{\mathrm{curr}}^{\delta, <q}} \right).$$

Note that the only difference between (1) and (2) is the source of the additional $(key, value)$ pairs: $p^\delta$. This, in particular, implies that all the operations with respect to the previous context are differentiable.

The number of different documents is equal to $b_S$ (the batch size, i.e. each document has a separate index in the batch). Assume that document $\delta$ has index $i$. We include into $p^\delta$ all tokens from $C_{prev}$ with the batch indices in $\{i, (i + 1) \mod b_s, \ldots, (i + d - 1) \mod b_s\}$.

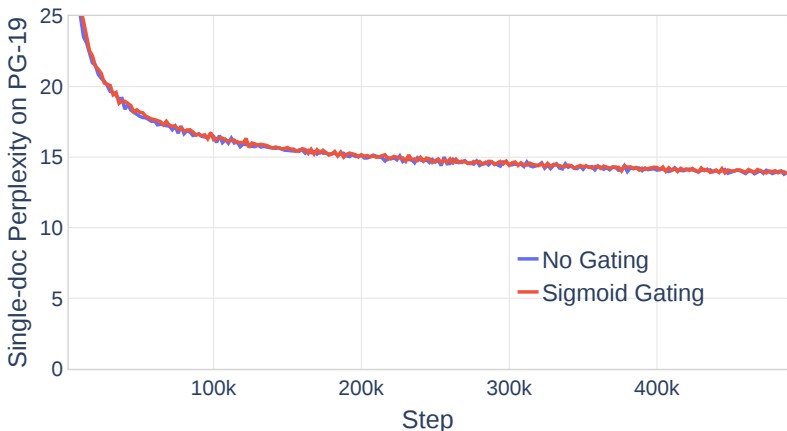

Figure 4: Perplexity (on the test set) during training on PG-19. We train two Memorizing Transformer models [Wu et al., 2022], one with original gating and one without (i.e. memory attention shared with local attention as described in (1)). We use the single-doc setting with 16k memory.

## B.4 Qualitative analysis

Table 7 provides a brief qualitative analysis of FoT. It shows that the model can handle distractions and retrieve the parts of the character name from the multi-document memory in the PG-19 task dataset and appropriate definitions from the large dictionary (dictionary lookup task).

## B.5 Memorizing Transformer

The Focused Transformer shares many similarities with the Memorizing Transformer [Wu et al., 2022]. In this section, we summarize the differences between those two models.

**Training** The key difference between these two methods lies in the training procedure. Our method uses *crossbatch*, see Section B.3, which, in a nutshell, is the standard transformer training objective, but we additionally attend to tokens from the previous context window, both from the same and different documents, see Appendix B.3 for details. The Memorizing Transformer trains on tokens retrieved from the same document (it was envisioned for single-doc).

This has a few important consequences:

- FoT does not use memory during training, while MT does. This may result in faster training; moreover, FoT always uses the most up-to-date values, while MT uses the values from memory, which may be outdated.

- FoT is differentiable through all $(key, value)$ pairs, while MT does not differentiate through the retrieved tokens. We argue that this is key for joint training of well-structured key, value, and query embeddings and, consequently, good model performance.

- FoT does not require long documents in the training set, while MT does in order to capture long dependencies in memory. This is practically important, as many popular datasets consist of short documents.

We speculate that there may be benefits in blending these two approaches. One can, for example, argue that MT is better at providing 'hard' negatives for the model. We provide a proof-of-concept experiment in Appendix C.3, and leave this for future work.

**Inference** Both models use a very similar memory attention layer. The difference is how the retrieved $(key, value)$ pairs are integrated. FoT treats the retrieved information in the same way as the local context. MT uses a gating mechanism. Details are provided in Section B.2.

Table 7: Example of elements retrieved by kNN search along with their scores on PG-19 and dictionary lookup task. The first column shows the token associated with a particular query (bolded) along with fragments of its context. The second column shows tokens associated with the keys retrieved by kNN for this query. The kNN score shows what fraction of the attention mass dedicated to retrieved keys corresponds to a particular key. The Focus score is calculated by taking the key along with 32 preceding and 32 following keys, calculating attention weights for them, and checking what fraction of attention the retrieved key gets. In the PG-19 setting the model was equipped with the memory of size 4096 spanning across parts of 8 documents. In the dictionary lookup setting the model was provided with memory of size $16M$.

| Text | kNN Results | kNN Score | Focus Score |
|---|---|---|---|
| **PG-19** | | | |
| S **HE** BA takes the gold-rimmed pince-nez which hangs upon THE DEAN'S waistcoat and places it before his eyes | Then if we're here with the closed carriage at ten–! [They go together into the library. DARBEY. [To SHE **BA** .] | 0.69 | 0.99 |
| | Oh! [He sinks on to the settee with a vacant stare, his arms hanging helplessly. DAR-BEY. [To SHE **BA** .] There–now his career is a burden to him! | 0.23 | 0.92 |
| | Papa! SHE **BA** . Papsey! [THE DEAN rouses himself, discovers his children and removes his hat. | 0.025 | 0.92 |
| THE DEAN gives DAR **BE** Y a severe look, and with an important cough walks into the Library. The men and the girls speak in undertones. | Then if we're here with the closed carriage at ten–! [They go together into the library. DARBE **Y** . [To SHEBA.] | 0.71 | 0.99 |
| | Oh! [He sinks on to the settee with a vacant stare, his arms hanging helplessly. DARBE **Y** . [To SHEBA.] There–now his career is a burden to him! | 0.19 | 0.99 |
| | Oh, Salome! Papa! Papa! TARVER. The Dean? DARBE **Y** . The Dean! | 0.08 | 0.99 |
| **Dictionary Lookup Task** | | | |
| <q> 14 42 23 38 <v> 40 41 05 56 | <k> 14 42 23 38 <v> 40 41 05 56 | 0.96 | 0.99 |
| <q> 30 55 07 23 <v> 36 17 26 63 | <k> 30 55 07 23 <v> 36 17 26 63 | 0.88 | 1.0 |
| <q> 10 41 26 39 <v> 48 38 11 24 | <k> 10 41 26 39 <v> 48 38 11 24 | 0.87 | 0.99 |

# C   Ablations

In this section, we focus on two key properties of crossbatch training procedure: differentiability and the inclusion of negatives. We also discuss the relation to Memorizing Transformer in terms of the training protocol and memory integration. We refer to Appendix B.5 for a detailed technical description of differences between FoT and Memorizing Transformer.

## C.1   Impact of differentiable keys and values

We compare FoT to Memorizing Transformer, which uses a non-differentiable memory of keys and values during training. In the multi-doc experiment presented in Figure 5, both MT and FoT are trained with local context of $512$. We observe that FoT is significantly better when the context is expanded during inference, which confirms that differentiable keys and values are beneficial.

Table 8: Perplexity on PG-19 in the single-doc setting for various local context lengths during training. In these experiments, we used the same context length both during training and evaluation.

| Context Length | FoT d=1 | MT |
|---|---|---|
| 512 | 14.18 | 14.68 |
| 1024 | 14.17 | 14.46 |
| 2048 | 14.11 | 14.43 |

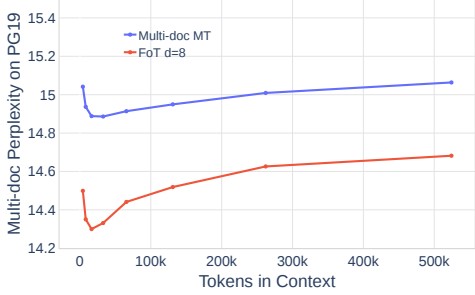

Figure 5: Perplexity on PG-19 in the multi-doc setting. Both FoT and Multi-doc MT were trained with local context of size 512. During training, Multi-doc MT utilized memory of size 4096 shared across 8 documents. Differentiability of keys and values results in better perplexity.

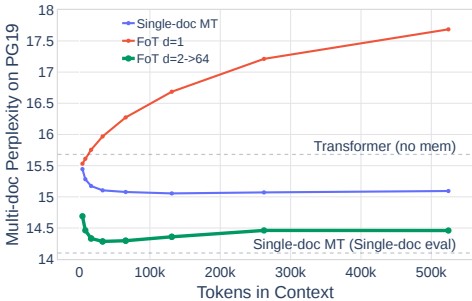

Figure 6: Importance of negatives in the multi-document setting. We compare FoT trained with $d = 1$ to the one that started with $d = 2$ and later switched to $d = 64$. For additional comparison, we show the performance of MT trained with the memory of size $16K$.

We also check whether differentiable keys and values can improve the performance in the single-doc setting. For this, we compare FoT with $d = 1$ to MT with memory consisting of the previous local context. Table 8 confirms that differentiable keys and values can also help in this scenario.

## C.2 Importance of negatives

We reaffirm the importance of negatives in a multi-document setting. In previous experiments in Figure 3, we already observed that increasing the number of negatives (i.e., increasing $d$) results in more attention mass being dedicated to relevant tokens. In Figure 6, we additionally show that the lack of negatives in training ($d = 1$) results in a significant deterioration in model perplexity when the context length grows. This confirms that both using negatives and differentiability are important for FoT to work well.

## C.3 Relation to Memorizing Transformer

Memorizing Transformer Wu et al. [2022] is closely related to our method. The two key differences are 1) the training protocol and 2) how the memory is integrated into the model. In this section, we provide additional insights into these differences.

**Training protocol** In the previous sections, we have discussed the benefits of the crossbatch training, namely using the contrastive-inspired objective and backpropagating through the previous context. A potential advantage of the MT approach is that it is exposed to the whole memory during training (instead of just the previous context). We performed a proof-of-concept experiment combining the two approaches to explore this further. Namely, we trained the model for 499k steps using crossbatch and fine-tuned it with the MT objective for 1k steps. Interestingly, we observed a significant improvement compared to the MT training with the same step budget, see Figure 7. We believe there is further room to explore various training protocols combining the best of both worlds.

**Memory integration** FoT uses a simple memory integration approach where the $(key, value)$ pairs retrieved by kNN lookup are treated the same way as the local context. In contrast, MT uses a gating mechanism, a weighted average of the memory, and local values; see details in Appendix B.2. We evaluated both approaches and found no difference in performance between these two memory integration methods. However, we decided to use our approach because it does not require

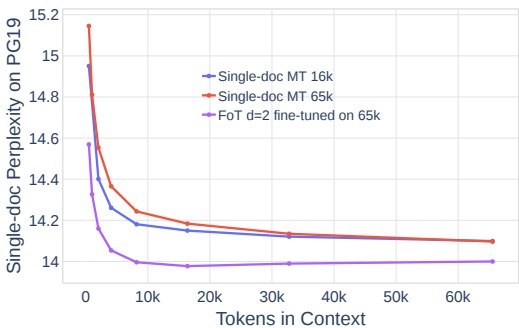

Figure 7: Single-doc eval of FoT finetuned for 1k steps on non differentiable memory. It achieves lower perplexity than MT, which has access to this memory for the whole training (500k steps).

any architectural changes (and thus makes fine-tuning existing models easy). For these reasons, we recommend using it. We speculate that the reason why the gating is not needed in FoT is another benefit of the fact that the crossbatch training backpropagates through the $(key, value)$ pairs from the previous context $C_{prev}$ in contrast to MT that cannot backpropagate there and needs to rely on local context when computing gradients for keys and values. Another reason might be the fact that $C_{prev}$ is embedded for each batch, and thus staleness (see [Wu et al., 2022, Section 3.2]) is avoided.

## D  Additional figures

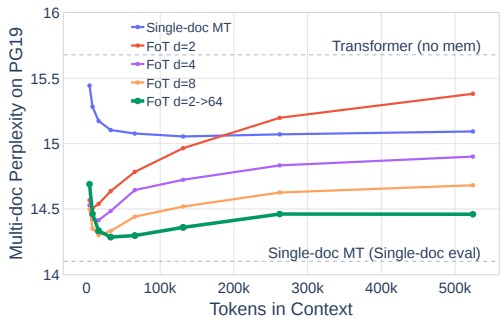

Figure 8: Perplexity in the multi-doc setting. FoT was trained with local context of size $512$ and different $d$. FoT 2->64 started with $d = 2$ and then switched to $d = 64$. Single-doc MT was trained with a memory size of $16K$. As we increase the memory size, the number of distractions (irrelevant keys) increases, making the task harder. Single-doc MT evaluated in the single-doc setting is a soft lower bound since it lacks distractions.

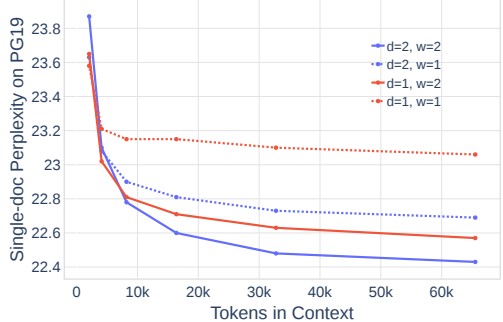

Figure 9: Zero-shot performance on PG19 of FoT pretrained on C4. Model fine-tuned with the crossbatch dimension $d = 2$ outperforms the one with $d = 1$. Using the double ($w = 2$) training context of 2048 is beneficial.

# E Hyperparameters

Table 9 shows hyperparameters used in our experiments. We used context length $512$ unless stated otherwise. In Appendix F, Section 5.3, Section 5.5, we use the total batch size of $32K$ tokens. In Section 5.2 and Section 5.4, the total batch size is $128K$ tokens.

Table 9: Hyperparameters for different model sizes. The batch size is given in number of tokens. For the $37M$ model local context length was 256 for FoT and 512 for the baseline.

| Hyperparameter | Value | |
|---|---|---|
| | **Common** | |
| #Layers | 12 | |
| Index of memory attention layer ($l$) | 8 | |
| Optimizer | AdaFactor | |
| Learning rate schedule | Inverse Square Root | |
| Warmup steps | 1000 | |
| $\beta_1$ | 0.9 | |
| | **Model-specific** | |
| #Params | $37M$ | $184M$ |
| Max learning rate | 0.02 | 0.01 |
| Min learning rate | 0.01 | 0.0005 |
| Embedding dim | 512 | 1024 |
| Head dim | 64 | 128 |
| #Heads | 8 | 8 |
| FeedForward dim | 2048 | 4096 |
| Local context length | 256/512 | 512 |
| Batch size | $32K/64K$ | $32K$ |
| #Number of training steps | $5k$ | $500k$ |

For the experiments described in Section 5.3 and Section 5.5 we performed the following hyperparameter sweeps:

- Learning rate: $\{1e{-}2, 5e{-}3, 3e{-}3, 1e{-}3\}$, chosen: $1e{-}2$,
- Batch size: $\{8K, 16K, 32K\}$, chosen: $32K$.

For the dictionary lookup task (Appendix F) we checked the following hyperparameters:

- Learning rate: $\{4e{-}2, 2e{-}2, 1e{-}2, 5e{-}3, 3e{-}3\}$, chosen: $2e{-}2$,
- Number of dictionary tokens in training step: $\{16K, 32K\}$, chosen: $32K$. Note that during the training number of document tokens dedicated to the dictionary is the same as the number of tokens dedicated to questions.

For most of the other hyperparameter choices, we followed [Wu et al., 2022], to provide a fair comparison.

## E.1 Schedule of $d$

In Sections 5.3, 5.4 and 5.5 for models with $d \in \{1, 2, 4, 8\}$ we used constant schedule, and for models with $d = 64$ we trained with $d = 2$ for $450k$ steps and switched to $d = 64$ for the final $50k$

steps. In Appendix F we trained with $d = 1$ until the model reached $98\%$ accuracy and then we switched to $d = 128$. For the $184M$ model in Section 5.2, we randomly sampled $d$ from $\{2, 128\}$ in each training step.

## F  Dictionary lookup task

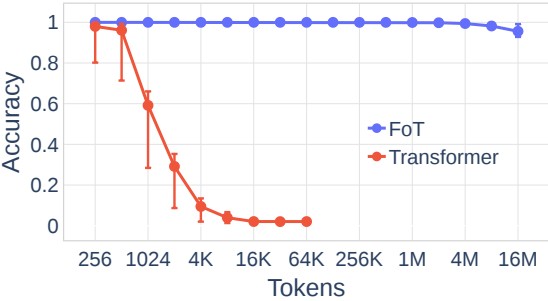

Figure 10: Accuracy vs number of dictionary tokens in a dictionary look-up task. The task format is as follows: `<k>`$k_1$`<v>`$v_1$`<k>`$k_2$`<v>`$v_2$...`<k>`$k_n$`<v>`$v_n$`<q>`$k_i$`<v>`$v_i$..., where a dictionary is provided, followed by queries on randomly selected keys. Accuracy is determined by measuring the predicted values $v_i$ after `<q>` tokens. Models were trained on examples containing $512$ tokens and evaluated with an extended context length. FoT demonstrates high accuracy even when the memory size is large. The baseline transformer fails already for $16K$ tokens. Error bars represent the minimum and maximum on 10 seeds.

We propose a dictionary lookup task to test whether the model trained using our crossbatch method can utilize a large memory database. Documents in this task are split into two parts. The first part defines keys and their associated values using the records of the format:

$$\texttt{<k>}, k_1, k_2, k_3, k_4, \texttt{<v>}, v_1, v_2, v_3, v_4,$$

where `<k>` is a special token that denotes the beginning of the defining sequence,

The second part consists of queries about the values associated with the previously defined keys. The queries are in the following format:

$$\texttt{<q>}, k_1, k_2, k_3, k_4, \texttt{<v>}, v_1, v_2, v_3, v_4,$$

where `<q>` is a special token that denotes the beginning of the query. We mask the loss so that for such a question, only $v_1, v_2, v_3, v_4$ are included. We use a vocabulary of $64$ tokens, keys and values are described using $4$ tokens.

During training, we use documents comprising $512$ tokens. The first half of each document consists of definitions, whereas the second one consists of questions. For FoT, we use a local context of $256$, thus the model needs to use the memory attention layer to answer the questions correctly. We start with $d = 1$ and increase to $d = 128$ as soon as the model is able to reach $98\%$ training accuracy. During the inference, we use $k = 32$ (the number of keys retrieved by kNN). As a baseline, we use a standard transformer model trained with the context length of $512$. In evaluation, we test different local context lengths, which quickly leads to very poor results.

In evaluation, we use longer documents but make only the last $256$ tokens correspond to questions. That is, as the context gets bigger (the token axis on Figure 10), the number of definitions increases, but the number of queries remains the same.

## G  FoT fine-tuning

For comparison in Table 3, our model is pre-trained for $100k$ steps with a total batch size of 128 ($128K$ tokens per step, with $1024$ local context). Then we fine-tune both FoT and baselines for additional $10k$ steps with the same batch size. When fine-tuning FoT, we randomly sample $d$ from $\{2, 128\}$ in each training step to prevent the model from overfitting to a large additional context length during training.

# H    Datasets

Section 5.1 outlines essential details concerning the PG-19 and arXiv datasets employed in this study. Now, we will present details about the remaining datasets:

**GitHub**   We obtained a large corpus of permissively licensed Github repositories using BigQuery. By filtering for specific file extensions (C, C++, Java, Python, Go, and TypeScript), we captured individual source code files that are often short but have numerous dependencies and cross-references within the repository. To preserve the structure while shuffling the files and subdirectories in a random order, we concatenated all the files within each repository, treating subdirectories as a unit, similarly to Wu et al. [2022].

**Isabelle**   The Isabelle corpus comprises formal mathematical proofs in the form of theories written in a formal language. We combined theories from The Archive of Formal Proofs (from October 2021) [3] and the Isabelle standard library to create a corpus of theories licensed as open source. Each theory focuses on topics like foundational logic, advanced analysis, algebra, or cryptography and consists of multiple files containing proofs. Similar to the GitHub corpus, the files within each theory are concatenated into a single document. However, unlike the Github corpus, we arrange the files based on their import dependencies, ensuring that later files can utilize sub-theorems proven in earlier files.

# I    Hardware and technical details

We used TPU virtual machines from the Google Cloud Platform (GCP). Each TPU virtual machine has 8 TPUv2 / TPUv3 cores totaling $64$GB / $128$GB of device memory, 96 CPU cores, and over 300GB of RAM. In larger-scale experiments (Section 5.2) we used machines with 32 TPUv3 cores. For training the LONGLLAMA checkpoints, a TPUv3-128 pod provided by the TPU Research Cloud was used, which we gratefully acknowledge.

# J    Randomness

To evaluate the significance of our results, we conducted multiple runs for selected experiments in our study. In Figure 10, we calculate error bars, showing the minimum and maximum value over 10 runs of the same experiment. For the arXiv baseline experiment in Appendix K, we performed three runs with different random seeds and calculated their standard deviation, which is equal to $0.002$ perplexity. However, due to resource constraints, we were unable to conduct multiple runs for all experiments. Our preliminary findings indicate that the observed variance was minimal compared to the impact observed from other factors under investigation.

For the calculation of test perplexities, we used $1M$ tokens.

# K    Additional experimental results

This section presents additional empirical results, providing a detailed comparison of FoT with the Memorizing Transformer [Wu et al., 2022] baseline. Both models are trained for the same number of $500k$ steps with local context of $2K$ and evaluated on the arXiv dataset in the single-document setup, following [Wu et al., 2022]. In particular, we study how models trained with a given context length perform when evaluated with different context lengths. These experiments differ from those in Section 5.2, as the models were both trained and evaluated on the same dataset (arXiv), unlike the C4 training and zero-shot evaluation done in Section 5.2.

The MT baseline in Table 10 with a memory length of $2K$ struggles to utilize additional context beyond $32K$ tokens effectively. The model trained with $8K$ memory performs significantly better when evaluated with longer contexts, showing further perplexity gains at $64K$ tokens. We observe diminishing returns when scaling up the training memory length to $16K$ tokens and beyond.

Using the same setup, we study the performance of FoT while varying $d$ and $w$ configurations, similarly to Section 5.4, see Table 11. Parameter values $w = 1$ and $w = 2$ correspond to additional

---

[3]https://www.isa-afp.org

context lengths of $2K$ and $4K$, respectively. In an apples-to-apples comparison to MT with $2K$ additional context length, FoT outperforms the MT baseline, which shows the importance of trainable keys and values (see also Section C.1). Moreover, we confirm the findings from Section 5.4 that $d = 2$ works significantly better than $d = 1$ in all settings. Our best configuration achieves 2.148 perplexity with $4K$ additional context during training, compared to 2.164 of MT with $16K$ additional context.

Table 10: Memorizing Transformer: arXiv perplexity values for different training memory lengths and evaluation context sizes

| Eval Context | Training Memory Length | | | |
|:---:|:---:|:---:|:---:|:---:|
| | 2k | 8k | 16k | 32k |
| $4K$ | 2.309 | 2.334 | 2.348 | 2.365 |
| $8K$ | 2.242 | 2.244 | 2.252 | 2.265 |
| $16K$ | 2.215 | 2.206 | 2.206 | 2.215 |
| $32K$ | 2.199 | 2.178 | 2.177 | 2.181 |
| $64K$ | 2.195 | 2.169 | 2.166 | 2.168 |
| $128K$ | 2.195 | 2.168 | **2.164** | 2.166 |

Table 11: FoT: arXiv perplexity values for different parameter combinations and evaluation context sizes

| Evaluation Context | Parameter Combinations (w, d), Training Memory Length | | | |
|:---:|:---:|:---:|:---:|:---:|
| | (1, 1), 2048 | (1, 2), 2048 | (2, 1), 4096 | (2, 2), 4096 |
| $4K$ | 2.292 | 2.299 | 2.305 | 2.309 |
| $8K$ | 2.229 | 2.224 | 2.214 | 2.217 |
| $16K$ | 2.206 | 2.194 | 2.178 | 2.178 |
| $32K$ | 2.192 | 2.176 | 2.159 | 2.156 |
| $64K$ | 2.187 | 2.171 | 2.152 | 2.149 |
| $128K$ | 2.187 | **2.171** | 2.152 | **2.148** |

# L  Code

In Listing 2, we show the FoTs attention code (i.e., the code for the memory attention layers and crossbatch training), see Section 3, Appendix B.2, Appendix B.3. We note that the changes to the code are small; they are localized to the memory layer (the other layers follow the standard transformer protocol) and do not require any new trainable parameters.

Listing 2: Memory attention: Let $\ell$ be a memory attention layer. During the training, we make $\ell$ attend to the $(key, value)$ pairs from the local context, previous local context, and $d - 1$ contexts coming from other documents. During the inference, queries from $\ell$ attend to the local context and $k$ nearest neighbors retrieved from memory. For simplicity, we provide the code for one head and assume that the crossbatch dimension $d$ is equal to the batch size.

```python
def mem_attn_layer(Ql, Kl, Vl, Cl, Km, Vm, Kp, Vp, attn_scf, mode):
  """Attention mechanism for crossbatch and memory attention
  Args:
    Ql, Kl, Vl: tensors of shape [batch, ctx_len, dim]
                with queries, keys and values from the local context
    Km, Vm:     tensors of shape [batch, ctx_len, k, dim]
                with  k most matching memory keys for
                each query from Ql along with associated values
    Kp, Vp:     tensors of shape [batch, ctx_len, dim] with
                keys and values from the previous context
    attn_scf :  a scale factor used before softmax
    mode:       either training or inference
  Returns:
    y: a vector with shape [batch, ctx_len, dim]
  """
  # attention to the local context
  local_attention = jnp.einsum("bqd,bkd ->bqk", Ql, Kl)
  local_attention *= attn_scf
  local_attention = apply_causal_mask(local_attention)

  if mode == "train":
    # In train mode, we additionally use previous context
    # and batch-1 contexts from other documents.
    prev_attention = jnp.einsum("bqd,ckd ->bqck", Ql, Kp)
    shape = prev_attention.shape
    additional_attention = prev_attention.reshape(shape[:-2] + (-1,))
  elif mode == "inference":
    # In the inference mode, we additionally use nearest neighbors
    # retrieved from memory. We retrieve k (key, value)
    # pairs for each query.
    memory_attention = jnp.einsum("bqd,bqnd -> bqn", Ql, Km)
    additional_attention = memory_attention
  else:
    raise Exception("Only train and inference modes are supported")

  additional_attention *= attn_scf

  # We merge the raw attention scores and calculate the softmax
  combined_attention = jnp.concatenate([local_attention,
                                        additional_attention],
                                       axis=-1)
  combined_weights = jax.nn.softmax(combined_attention, axis=-1)

  ctx_len = Ql.shape[1]
  local_weights = combined_weights[..., :ctx_len]
  additional_weights = combined_weights[..., ctx_len:]

  y = jnp.einsum("bqk,bkd -> bqd", local_weights, Vl)

  if mode == "train":
    prev_weights =  additional_weights
    shape = prev_weights.shape
    prev_weights = prev_weights.reshape(shape[:-1] + (-1, ctx_len))
    y += jnp.einsum("bqck,ckd -> bqd", prev_weights, Vp)
  else:
    memory_weights = additional_weights
    y += jnp.einsum("bqn,bqnd -> bqd", memory_weights, Vm)
  return y
```

