# OpenReview forum: "Focused Transformer: Contrastive Training for Context Scaling"
_NeurIPS.cc/2023/Conference — NeurIPS 2023 poster_

### Official Review · Reviewer_GWpw · 2023-06-26

**Soundness:** 4 excellent
**Presentation:** 4 excellent
**Contribution:** 3 good
**Rating:** 8
**Confidence:** 4

**Summary:**

This paper proposes an improved training approach for memory-augmented Transformers on language modeling tasks. In particular, the authors identify the distraction issue in memory-augmented models, whereby the attention mechanism tends to focus on irrelevant contexts in the regime of long sequences. To address this concern, the authors propose cross-batch training, which is inspired by contrastive learning and exposes the attention mechanism to process both relevant and irrelevant documents. Extensive experiments are conducted to validate that the proposed cross-batch training can effectively shape the key-value space, mitigate the distraction issue, and successfully extend the model context size.

**Strengths:**

- The paper is clearly written and well structured.
- The identified distraction issue is compelling and makes a valuable contribution to the long-context research community, highlighting the need for more effective attention mechanisms or retrieving techniques.
- The method is straightforward to use and demonstrates its significant effectiveness in identifying relevant information.

**Weaknesses:**

My main concern is about the increased training computational costs. The computational complexity for the memory-augmented attention layer increases from $O(bn^2)$ to $O(b d n^2)$, where $b$ represents the batch size, $d$ is the number of cross documents, and $n$ is the sequence length. Employing a large $d$ might introduce a significant computation overhead.

**Questions:**

1. How does the proposed technique compare to conventional packing techniques commonly used during pretraining (e.g., concatenating multiple documents together, possibly interleaved with [EOS] tokens)? This technique is mostly used to reduce padding but also generates a sequence with context from diverse documents.
2. How well does the proposed model converge in comparison to vanilla transformers? Intuitively, as the attention mechanism in FoT is exposed to more irrelevant information, the learning process might slow down.
3. How can one determine the optimal transformer layer for memory augmentation? Does allowing more transformer layers access to the memory aid in the retrieval of relevant information?

**Limitations:**

The authors have adequately addressed the limitations.

---

> ### Author Rebuttal · Authors · 2023-08-09
>
> We thank the reviewer for a thoughtful review.
>
> **Regarding the computational cost**. We thank you for raising this important concern, which we have added to the limitation section. We also note that two factors mitigate this issue to some extent. As you noted, the increased cost occurs only in the memory layer. Second, FoT exhibits some context extrapolation, which might allow using smaller $n$ in training. Accordingly, we managed to fine-tune OpenLLaMA models using $d \leq 8$
>
> Regarding the questions:
>
> **Q1:**  We thank the Reviewer for proposing an interesting baseline. To answer the question, we fine-tune a vanilla OpenLLaMA model on sequences of length $4096$ (original seq. len $2048$), which we consider as just a standard “data packing" baseline, and compare it to a FoT model trained on exactly the same data packed in the same way for 1B tokens. For clarity, we outline the following architectural differences between the baseline and FoT:
> * Additional context beyond $2048$ tokens is used in just a subset of layers
> * FoT does not use positional encodings in memory layers beyond its original context window ($2048$)
>
> The results are as follows:
>
> |Context/Setup | TREC: baseline $\pm 1.0$  $~~$ | TREC: FoT $\pm 1.0$  $~~$  | WebQS: baseline $\pm 0.1$  $~~$  | WebQS: FoT $\pm 0.1$  $~~$  |
> | - | - | - | - | - |
> | 2K | 52.8 | 55.6 | 20.7 | 20.8 |
> | 4K | 57.2 | 60.9 | 18.7 | 21.0 |
>
> We observe accuracy improvements when more few-shot demonstrations are provided in the extended context (from 2K used by OpenLLaMA to 4K used in our fine-tuning). On TREC, the gains from additional context are significant for both models. Our method presents better data efficiency than the baseline.
>
> **Q2:**  Starting with a large $d$ (crossbatch dimension) may slow down the process and result in the memory layer being ignored by the model. We have not seen any such problems when starting with a smaller value of $d\leq8$. See the plot with the training loss comparison in the attached pdf.
>
> **Q3:** Due to the limited resources, we have followed the choice of Memorizing Transofrmer (MT) in picking the memory layer. We have also seen some additional gains from using multiple memory layers in our FoT fine-tuned OpenLLaMA models.
>
>
> If your concerns have been sufficiently addressed in our responses, we humbly seek your support for the paper. Should you have any further concerns or additional points to raise, we are eager to address them.

---

> > ### Comment · Reviewer_GWpw · 2023-08-16
> >
> > I thank the authors for their detailed responses. The clarification resolves most of my concerns, and it is nice to see the improvements due to FoT over naive sequence packing, which validates its benefits. The new experimental results on both short- and long-context tasks further strengthen the claims. My rating has been adjusted upward due to the greatly enhanced clarity.

---

> > > ### Author Response · Authors · 2023-08-16
> > > **Thank you**
> > >
> > > Thank you for supporting our work and, again, for useful suggestions.

---

### Official Review · Reviewer_fPf8 · 2023-07-04

**Soundness:** 4 excellent
**Presentation:** 3 good
**Contribution:** 4 excellent
**Rating:** 8
**Confidence:** 5

**Summary:**

This paper proposes the Focused Transformer (FoT), which modifies one layer of the Transformer model's Attention layer to Memory Attention, thus enabling the model to learn almost infinite length context without being limited by the constraints of local attention. Specifically, the paper also proposes a cross-batch training method that can adapt the model to any length context at a relatively low fine-tuning cost. Experimental results demonstrate the superiority of FoT on Long Context tasks.

**Strengths:**

1. By only retaining one layer of Global Attention (Memory Attention), both training and inference are efficient;

2. During the inference stage, it can flexibly follow the results of kNN retrieval;

3. The cross-batch training method is very effective, and the training cost is also relatively low;

4. The increase factor of the Context length for LLM reached 2^15!

**Weaknesses:**

1. With all due respect, I believe that the main text of the paper does not clearly introduce FoT and cross-batch training. I didn't understand it until I read the code in the appendix;

2. The inference stage needs to be combined with kNN, which is inconsistent with the training mode and may affect the upper limit of the model's ability;

3. There is a lack of comparison with other similar schemes for increasing Context length (such as Parallel Context Window).

**Questions:**

1. I don't quite understand why, during the cross-batch training stage, we need to distinguish between C_prev and C_curr? Isn't it better to treat all tokens across batches as negative learning directly?

2. In the inference stage, how are the K and V to be input into Memory Attention calculated after kNN?

3. After cross-batch training, will the model overfit long Context? Is its performance on short Context still good?

**Limitations:**

The authors have clearly and sufficiently explained the limitations of their research

---

> ### Author Rebuttal · Authors · 2023-08-09
>
> We thank the reviewer for the encouraging review.
>
> The description of the method outlines the general idea, and we admit that it might be hard to infer details from it. We think presenting the details in the text would be quite cumbersome; thus, we plan to include the shortened versions of the code in the main body of the paper. We hope this will be satisfactory.
>
>  **Regarding the kNN**, we consider kNN to be an approximation of the full dense attention. With such a perspective, there is no inconsistency. However, in practice, the approximation errors may impact the performance. We did not observe this in our experimental regime. We leave proper studies of this to future work. We also note that our fine-tuned versions of OpenLLaMA models use full dense attention instead of kNN, which we find performant and efficient enough. We also note that using kNN opens the possibility of using fast approximate indices (e.g., implemented in Faiss), which might be necessary for scaling the method. We have added this to future work.
>
>  **Regarding the comparison with Parallel Context Window**, we add the following text to related work.
> > Parallel Context Window introduces a method for extending the context of language models without training. They achieve this by embedding several context windows independently in parallel and allowing only a subset of tokens to attend to all windows. On the other hand, we fine-tune existing models and allow all tokens to attend to all previous tokens but only in a subset of layers. Our method also allows us to improve the structure of $(key, value)$ space of existing models.
>
> Regarding the questions:
> 1. We observed that it is important to have at least one positive example that brings additional related information to memory layers (for example, previous local context window $C_{prev}$). Otherwise, the model may learn to ignore memory layers.
> 2. For each query in a memory layer, we take $k$ most matching keys from memory and add them to the attention for this query. That is, each query will attend only to all keys that precede it in the local context and its own $k$ most matching keys from memory. In the non-kNN approach, each query attends to the whole memory and all keys that precede it in the local context. To calculate $k$ most matching keys, we use the inner product. Note that in models presented in the paper, we remove positional encodings in memory layers.
> 3. We have managed to fine-tune OpenLLaMA models so that they maintain the performance of the base models on short-context Language Model Evaluation Harness tasks and show improvements on long-context ones. For details, please refer to the table from the general response.
>
> We again thank you for an encouraging review. Should you have any further questions or concerns, we'd be happy to answer. We kindly ask to support our paper.

---

> > ### Comment · Reviewer_fPf8 · 2023-08-12
> >
> > Regarding question 2, I still don't quite understand, maybe I didn't express the question clearly, sorry.
> >
> > What I want to know is, after we have retrieved the topk contexts related to the query in some measure (at this point these contexts should be pure text), how should we encode these contexts so that they can be concatenated into the key sequence of Memory layer.

---

> > > ### Author Response · Authors · 2023-08-13
> > > **clarification**
> > >
> > > The granularity of the retrieval in our work is token-level, like in [1] (not passage/context-level, like, e.g., in [2]). For each query, we retrieve k keys and values that are vectors, not text. These keys and values are integrated via a kNN-attention mechanism similar to [1].
> > >
> > > For example, for $k = memorySize$ and a one-layer model, this is equivalent to extending the local context length and using standard attention instead of kNN (kNN with this k 'retrieves the whole memory'). To be more precise, the memory consists of $(key, value)$ pairs generated for each token in the chosen memory layer of the transformer model.
> > >
> > > We note that Appendix A.2. details the inference procedure (and differences with Memorizing Transformer [1]) using formulas. We could potentially move (more parts of) it to the main part. Likewise, we’d be happy to apply other suggestions if you think this would clarify the exposition.
> > >
> > > We thank you again for pushing this point; we’re determined to make it more clear.
> > >
> > > [1] Yuhuai Wu, Markus N. Rabe, DeLesley Hutchins, Christian Szegedy. Memorizing Transformers, 2022

---

### Official Review · Reviewer_v42b · 2023-07-06

**Soundness:** 2 fair
**Presentation:** 2 fair
**Contribution:** 2 fair
**Rating:** 6
**Confidence:** 4

**Summary:**

This work presents a modified method to train Transformers with memory. The memory is based on k-Nearest Neighbors (with exact match). The work is based on the Memorizing Transformer paper from Y. Wu et al. 2022.
The model assumes a transformer model with a memory attention (an attention layer that has inputs from both current sequence and memory output). The (k-NN) memory returns the top-k matches for a query, and these are included in the input to the memory attention. One difference is the removal of a gating mechanism that mixes between memory and local context, becoming all input to an attention layer.
In addition, contrastive learning is proposed as a method to enhance the model capability to differentiate relevant keys from irrelevant ones in the memory layer. This is achieved by including negative samples from other documents, applying a forward pass up to the memory attention layer, and using these key,value pairs as simulated responses from the k-NN memory. These samples are mixed with the previous context block from the current sample. This method is differentiable and allows to backprop through these positive and negative samples, requiring only a modification to the data pipeline. The authors suggest that the contrastive learning is needed do reduce a "distracting issue" caused in a multi-document setup.

Eventually, the authors show the value of their method with a set of experiments that compare mainly against the memorizing transformer. The results show that memory can extend the effective context length for language modeling at inference time, they can solve a copy task using memory, a pre-trained model can be fine-tuned to introduce the memory attention layer, improves performance in single and multiple document scenarios, and an ablation study with the importance of the negative samples and the differentiable keys.



**Strengths:**

The paper mixes the idea of negative sampling and a twist to the memorizing transformer model with a k-NN memory. The topic is of significance to this community.

* At inference time this model can increase the context length beyond the training length. Inheriting the original memorizing transformer property.
* It is possible to fine-tune from a pre-trained model with no memory to a model with external memory.
* The experiments show the value of the negative samples. The training methodology behind the idea is relatively simple.


**Weaknesses:**

The contribution of this work has limited novelty: both negative sampling (and contrastive learning [2]) and memorizing transformers have been proposed in the past for language modeling.

The clarity and explanations in the work could be improved. The work continuously omits important details or descriptions, sending the reader in most cases to the appendix. The references could make a better exploration of contrastive learning methods applied to language modeling (e.g., [2]). Please, see questions for additional details that may need clarifications.

Retrieval-augmented language models are able to utilize memory to gain information from multiple documents. The "distracting issue" described in the paper suggests that the memory attention mechanism gets distracted by keys from different documents. However, this seems to contradict the findings in previous work (retrieval methods) that earn performance using multiple documents.

The experiments are mainly limited to the memorizing transformer and the vanilla transformer. However, retrieval language models solve the same multi-document problem. Also the dataset chosen follow the memorizing transformers, failing to compare with existing benchmarks in the long context use case, like SCROLLS [1].

[1] Shaham et al., SCROLLS: Standardized CompaRison Over Long Language Sequences, 2022
[2] Jain et al., CONTRACLM: Contrastive Learning For Causal Language Model, 2022

**Questions:**

I would appreciate it if the authors could answer the following questions:

* Line 188: Why is layer 8 picked as the memory layer? How would someone choose the "right" layer for connecting the memory?
* What is the improvement of an existing language model (like OPT, LLaMA, etc.) when fine-tuned to introduce the memory attention layer?
* Why is your model not reaching perfect accuracy in the synthetic task? (Shallow) transformers are able to obtain and copy information. Is the same key related to multiple values in different documents in the dataset?
* What is stored in memory during the evaluation in Section 4.3? What is the data in the memory?
* What is the distance measure for k-NN?
* How is the number of neighbors $k$ chosen for the memory retrieval?



**Limitations:**

The authors address limitations of their work in Section 5.
Scaling (approximate) k-NNs and memory has been addressed for quite some time. See the NeurIPS 2021 competition on Billion Scale Approximate Nearest Neighbor Search.

---

> ### Author Rebuttal · Authors · 2023-08-09
>
> We thank the Reviewer for their thoughtful feedback. We acknowledge some deficiencies in the presentation. We focus on the long-context capabilities; the appropriate clarification is described in the general answer. In more detail, we aim for a single-stage method that can incorporate a large number of tokens directly in the model context (kNN attention can be used to approximate full attention). We observed that increasing context length naively gives worse results which is also confirmed e.g., in [1]. This is not in contradiction with the fact that the retrieval methods benefit from additional documents. The difference lies in the fact that they typically use a two-stage approach, with the retrieval part doing the hard job of extracting only a relatively small amount of tokens, which are efficiently processed within the standard context length [2]. We include this clarification in the paper. We also make a number of smaller adjustments to the paper, which hopefully make the paper easier to follow. If the Reviewer sees any specific issue, we'd be happy to address it.
>
> We acknowledge some issues with clarity. In the method description, we outline the general idea and admit that it might be hard to infer details from it. We think presenting the details in the text would be quite cumbersome. To amend the situation, we plan to include a shortened version of the pseudocode from the Appendix in the main body. The pseudocode has been found helpful by Rev #fPf8. Thus, we hope it will satisfactorily complement the description. If you see any other parts which require clarification, please let us know.
>
>
> Below we address the questions:
> 1.  As noted in the general response, due to the limited computational resources, we could not perform full hyperparameter sweeps. In particular, for the memory layer, we have followed the choice of Memorizing Transformers (MT) [3].
> 2.  Regarding the improvements in existing language models and benchmarking on additional long-context tasks, as noted in the general response, we present 3B and 7B models based on OpenLLaMA along with results on Qasper (SCROLLS benchmark), TREC, and WebQS where we show improved performance when the model is provided with additional context.
> 3. Regarding the performance on the synthetic task. Please note that the model is trained in a much shorter context than it is evaluated, which makes it out of distribution.
> 4. Regarding the memory content in Section 4.3 during evaluation - this is a single-doc memory; that is, in the additional context, we only store keys and values belonging to the currently processed document.
> 5. The distance measure for kNN is inner product.
> 6. We have tested values of $k\in \{32, 64, 128\}$ and observed small differences in performance. We add this information to Appendix.
>
> Thank you for pointing out [4]; we add the following description to the related work section:
> > CONTRACLM [4] applies contrastive losses at both the token and sequence levels during training to promote more uniformly distributed, isotropic representations. It is shown to enhance the discrimination of representations on textual semantic similarity benchmarks. While CONTRACLM focuses on improving the general expressiveness of representations, our work introduces contrastive-inspired techniques designed specifically for training the memory attention mechanism to handle longer context lengths. Nonetheless, exploring other contrastive learning objectives could be beneficial for further improving the memory key structure in future work.
>
>
> [1] Nelson F. Liu et al., Lost in the Middle: How Language Models Use Long Contexts, 2023
>
> [2] Sebastian Borgeaud et al., Improving language models by retrieving from trillions of tokens, 2021
>
> [3] Yuhuai Wu, Markus N. Rabe, DeLesley Hutchins, Christian Szegedy. Memorizing Transformers, 2022
>
> [4] Jain et al., CONTRACLM: Contrastive Learning For Causal Language Model, 2022
>
> We again thank the Reviewer for raising important issues. We hope that our answers are satisfactory. If not, we’d be happy to provide more details. Otherwise, we’d appreciate it if the Reviewer reconsidered the final score of our submission.

---

> > ### Comment · Reviewer_v42b · 2023-08-17
> >
> > I would like to thank the authors to take time to address my questions.
> > Accordingly, I have increased my score to Weak Accept.
> >
> > Thank you

---

### Official Review · Reviewer_Sux9 · 2023-07-06

**Soundness:** 3 good
**Presentation:** 3 good
**Contribution:** 2 fair
**Rating:** 5
**Confidence:** 4

**Summary:**

The paper proposes a key challenge (distraction issue) when extending the attention layer to external (key, value) pairs, either from previous context of the same document or from other documents. This paper proposes the Focused Transformer (FoT), a model that can utilize a long context by retrieving kNN (key, value) pairs from memory. The architecture of FoT is similar to Memorizing Transformer (how retrieved items are integrated is different), but the training technique is different. This paper proposes to use crossbatch to train the model, which simply includes both previous context of the same document and also contexts from different documents. So the model can learn how to distinguish between useful information and distraction.

**Strengths:**

* The paper focuses on an important and potentially impactful problem that extends the context window size of transformer models.
* The proposed model and the training method is straightforward and easy to implement.
* The experiments are fairly solid and have supported the points that the paper has raised.
* I also like the synthetic dictionary task as an evaluation task to test if a language model has the ability to attend to desirable context and gather necessary information.

**Weaknesses:**

1. One main weakness of this paper is that it is not clear to me if the proposed model is designed to incorporate long context or to incorporate external memory which may come from a large corpus.
As the paper includes both single-doc and multi-doc experiments, I assume it is the latter case. Based on this assumption, the paper identifies “distraction issue”, which essentially means the model attends to more other documents when considering more documents during inference.

I did not get the point here. If we allow the memory to contain items from other documents, we actually expect the model to extract useful information instead of treating them as “distraction”. When showing the problem of “distraction issue” (Fig 3), we only consider 64 documents, which is far from the real case where we want to use external corpus. Indeed, prior works have shown that using more external information can help the model to achieve better results instead hurting it, e.g., (Khandelwal et al., 2019), (Zhong et al., 2022).

While if the paper is mainly considering only the long-context cases, we also should question if the “distraction issue” exists during inference or not, because we can always control the model to only attend to a single document.

So, I don’t think the paper is well-motivated or well-positioned. The authors are encouraged to clearly state what testing situations they are addressing and re-evaluate the proposed issue in that situation.

2. The paper can be presented better. For example, it is never clear to me what “external memory” exactly means in the paper. Why “previous local context of the given document” is called positive and “contexts from unrelated documents” is called negative, given there are really no distinctions in the training objective (correct me if it is not the case)?

3. The proposed CrossBatch technique is based on including both previous contexts from the same document and contexts from different documents in the same training batch. This method is very similar to the data batching method proposed in (Zhong et al., 2022). The authors are encouraged to discuss the differences.

**Questions:**

* L144: why is there a positive and negative? Did you actually distinguish positive/negative in the contrastive loss?
* Table 2: why do you use token-level accuracy here but use perplexity elsewhere.
* How do you define external memory? Anything that is out of the original input can be called external.

**Limitations:**

The authors have well discussed the limitations of the research.

---

> ### Author Rebuttal · Authors · 2023-08-09
>
> We thank the reviewer for their constructive feedback.
>
> We admit deficiencies in clarity raised by the reviewer. We note that we focus on *the long-context capabilities*; see also the general response. In our experiments, we tested FoT in both single-doc and multi-doc scenarios to assess its potential usefulness. We found that FoT improves perplexity on single, long documents (see Section 4.5), and we believe this makes it applicable to generic long-context language modeling, which is strongly confirmed by our new experiments with large models. At the time of writing, we decided to keep some multi-doc experiments, e.g., to illustrate the distraction issue, which already impairs the model’s perplexity significantly at a relatively small scale (64 documents, see Figures 3,7). However, in retrospect, we recognize this might be confusing. To amend this, we apply the steps described in the general answer; in particular, we indicate the long context focus more explicitly.
>
> We also note that there are practical applications where the multi-document setting is well-motivated, particularly repository-level code generation. We hope that our method could be scaled up to open possibilities for handling the entire repositories of code in context (possibly ~1M tokens in large codebases), which we plan to attempt in future work.
>
>
> **Regarding the 'external memory'**. By this, we understand anything outside the local context window, i.e., anything that is accessed additionally by memory attention layers. This clarification has been added to the paper. To make this name less confusing, we changed it to 'additional context'.
>
> **Regarding 'positive and negative examples'**.
>
> Our method is inspired by contrastive learning in the way how data is presented to the model. We assume the model is presented with distractions (possibly irrelevant documents) in the training phase. The previous local context (from the current document) is mixed with contexts from other documents in the batch. Intuitively, this 'forces' the model to learn to differentiate between the positives (tokens from the current document, which are likely to be useful) from the negatives (tokens from other documents, which are unlikely to be useful). We note that this is not standard contrastive learning, as we do not have a separate contrastive loss. We only use the standard language modeling loss. We have added a clarification to the paper.
>
> **Regarding Table 2**. We agree that it is not the best way to compare the models, but we were constrained by pre-trained models (as different tokenizers are used, comparing perplexity is not informative). We only aim to show that we get better accuracy with more context available for a given single model, in contrast to comparing token-level accuracies between models with different tokenizers, which is inconclusive. A comment has been added to the caption.
>
>
>
> **Regarding [1, 2]**
> We first mention that our focus is different. As now clarified, we aim for a long context, while these papers are focused on retrieval from a large knowledge database. We have added a clarification to the related work section. On the technical level, [1] combines two probability distributions to predict the next token: one given by the model logits and the other created from retrieved pairs of (embedding, next token). Meanwhile, we extend the model context in a subset of attention layers, potentially allowing for reasoning within this extended context.
>
> We thank the Reviewer for raising the topic of the usefulness of other documents in the batch. It was observed that nearest neighbors language models (kNN-LM) display almost linear perplexity gains wrt. datastore size[3]. Due to practical limitations, we embed tokens on the order of magnitude of ~100K per training batch, and documents in the batch are randomly sampled from a large corpus, which means it is unlikely that they are related to each other. Therefore, we should not expect significant perplexity gains for kNN-LM in that setting either, as the training bach comprises approximately 0.1% of the datastore. Empirically, we show that extending the model's context length with attention instead of using kNN-LM leads to perplexity increase due to the aforementioned distraction issue. To the best of our knowledge, the distraction (perplexity increase) resulting from increasing attention context length hasn't been studied before.
>
> We agree with the Reviewer that TRIME [2] proposes a very similar objective inspired by contrastive learning, which is already mentioned in the related work section. The main difference is architectural: instead of attending to additional tokens in the memory layer, like in FoT, they combine probability distributions of the dense model and the retrieval database in the final layer, like [1]. Moreover, [2] focuses on retrieval from large databases, whereas our experiments mostly focus on long context. We have included this discussion in the related work section of the updated paper.
>
>
>
> **Regarding the distraction issue at the inference time**, giving the model multiple unrelated documents is an extreme case. The distraction issue could possibly occur in single-doc scenarios for long documents consisting of several chapters. Please note that despite alleviating the distraction issue, FoT allows training long-context models using short-context data and improves performance in single-doc cases (see Section 4.5).
>
>
> [1] Khandelwal et al., Generalization through Memorization: Nearest Neighbor Language Models, 2019
>
> [2] Zhong et al., Training Language Models with Memory Augmentation, 2022
>
> [3] Xu et al., Why do Nearest Neighbor Language Models Work?, 2023
>
> If our responses have adequately addressed your concerns, we kindly request your support and considerating of improving your score. If you have any further concerns or additional points to raise, we are eager to address them. Your feedback is valuable in enhancing the quality and impact of our research.

---

> > ### Author Response · Authors · 2023-08-21
> > **discussion**
> >
> > We hope that our explanations and changes are clear enough. If there is something else, we would be happy to address it in the remaining time of the rebuttal period.

---

> ### Comment · Reviewer_Sux9 · 2023-08-21
>
> I appreciate the authors' response to my review. I am glad that the response has addressed my primary concern (weakness 1) regarding this paper. I have increased my score.

---

### Author Rebuttal · Authors · 2023-08-09

We would like to thank you for all your valuable feedback, both positive and negative, which we believe will help us to improve the quality of our work.

We are delighted to note that the reviewers (Sux9, v42b, GWpw) prized the simplicity of our method and noted the potential impact (Sux9) of extending the context length (v42b, GWpw). Reviewer fPf8 noted the efficiency of FoT, and Sux9 prized the synthetic dictionary task.

Reviewers Sux9 and v42b raised important concerns about the scope and contributions of the paper, which are addressed below. Moreover, we would like to advertise new important experiments.

### New experiments with large models
In the period between the submission and the rebuttal, we secured additional compute resources, which let us confirm that our method is useful for much larger models. We believe that this significantly strengthens the paper. Specifically, we fine-tuned $3B$ and $7B$ OpenLLama models with our FoT objective. The resulting models exhibit advancements in tasks requiring long context. Following that, we extend the contribution list accordingly.

Below we shortly summarise the properties of our models. We would be happy to provide more details here if needed. Otherwise, we present them in an additional section in the paper. Specifically, our new models:
1. exhibit long-context capabilities on downstream tasks (see tables below),
2. retain the performance of the original models on short-context tasks,
3. are more compute and memory efficient at inference, compared to vanilla Transformers with the same effective context length,
4. have some context extrapolation capabilities. We illustrate that our models manage a 256k context length for passkey retrieval task from [1]  even though being trained on 8k context. **(see pdf)**.

Ad 1. Our model exhibit performance gains from additional in-context few shot examples on TREC question classification [2, 3] and WebQS question answering [4]. What is more, it shows improvements in F1 score on Qasper (Question Answering over Scientific Research Papers) task [5], which is a part of SCROLLS [6].







| Context/Setup | TREC: FoT fine-tuned OpenLLaMA 3B | TREC: FoT fine-tuned OpenLLaMA 7B | WebQS: FoT fine-tuned OpenLLaMA 3B | WebQS: FoT fine-tuned OpenLLaMA 7B |
|---------|---------------------------------|---------------------------------|----------------------------------|----------------------------------|
| 2K  |          67.0                   |          63.2                   |           21.2                   |            25.5                  |
|  4K    |          71.6                   |          72.7                   |           21.4                   |            26.4                  |
|   6k    |          72.9                   |          74.9                   |           22.2                   |            27.2                  |
| 8K      |          73.3                   |          75.9                   |           22.4                   |            27.7                  |

For Qasper, we used the implementation from Language Model Evaluation Harness and observed that our model 3B model benefits from context increase. Below we provide zero-shot results. Note that LongChat 7B [7] was instruction fine-tuned.

|Context length | OpenLLaMA 3B | FoT fine-tuned OpenLLaMA 3B | LLaMA 7B | LongChat 7B |
| - | - | - | - | - |
| 2K | 18.7 | 18.7 | 18.7 | 19.4 |
| 4K | - | 20.7 | - | 21.2 |
| 6K | - | 23.2 | - | 25.0 |
| 8K | - | 26.6 | - | 28.8 |


Ad 2. Our fine-tuned OpenLLaMA models maintain the performance on the standard suite of short-context  tasks from Language Model Evaluation Harness (we use the same collection of tasks as OpenLLaMA and provide the average scores)

|Model | OpenLLaMA 3B | FoT fine-tuned OpenLLaMA 3B | OpenLLaMA 7B | FoT finetuned OpenLLaMA 7B|
| - | -| - | - | - |
|Average score | 0.53|	0.53 |	0.55 |	0.55|


### Scope and contributions of the paper

To clarify, *our paper focuses on the long-context capabilities*. We agree that the current writing is somewhat unclear. We have identified the following issues which might have caused the confusion:

- We now stress that handling large external databases was the initial motivation of FoT, which was later changed to long-context.
- We used the term 'external memory', which we now change to 'additional context'.
- Memorizing Transformer, on which we base our method, is framed as a retrieval method. We now explicitly state in the related work section that despite these similarities, our aim is different. Moreover, we amend the related work to include more long-context papers.
- We include new long context tasks (see above). We keep the multi-doc experiments for illustrative purposes. However, we make explicit that the focus is on the long context.

We thank the reviewers for pinpointing this clarity issue. We hope that the above changes will address the concerns. We would be happy to make further adjustments if the reviewers find it useful.


### Tuning and hyperparamters

Reviewers (V42b, GWpw) raised questions about hyperparameters (e.g. the memory layers used). We note that some of the choices were educated guesses, as due to extreme computational cost, we could not perform a full hyperparameter search. For example, this was the case for the memory layer we based on the findings from Memorizing Transformer. This information is now added as a limitation.


[1] A. Mohtashami, et al. Landmark Attention: Random-Access Infinite Context Length for Transformers.

[2] Li, Xin et al. Learning Question Classifiers.

[3] E. Hovy, et al. Toward semantics-based answer pinpointing.

[4] J. Berant, et al. Semantic Parsing on Freebase from Question-Answer Pairs.

[5] P. Dasigi, et al. A Dataset of Information-Seeking Questions and Answers Anchored in Research Papers.

[6] U. Shaham, et al. SCROLLS: Standardized CompaRison Over Long Language Sequences.

[7] D. Li*, et al. How Long Can Open-Source LLMs Truly Promise on Context Length?

---

### Decision · Program_Chairs · 2023-09-21

**Decision:**

Accept (poster)

**Comment:**

The paper proposes the Focused Transformer (FoT) to addresses the limitations of local attention in Transformers by incorporating a Memory Attention layer, enabling it to learn from extensive context while mitigating distractions. This architecture, reminiscent of the Memorizing Transformer, utilizes a k-Nearest Neighbors (k-NN) memory retrieval mechanism to distinguish between relevant and irrelevant information. FoT's unique crossbatch training approach trains the model with context from both the same and different documents, enhancing its ability to discern valuable insights. Experimental results underscore FoT's success in handling long-context tasks and improving performance across various scenarios, positioning it as a promising solution for extending context and managing distractions through innovative attention mechanisms and training techniques.

All reviewers appreciate this work and the discussion between authors and reviewers has addressed most of reviewers' concerns. Therefore, I am happy to see this paper accepted at the conference.